# Determinants and disparities in skilled birth attendants during childbirth in Bangladesh: A study of machine learning and decomposition analysis

Rafayet Rahman Ridoy📷, Rehana Parvin📷°*, Tofayel Ahmed°, Tahira Mahbub°

Department of Statistics, International University of Business Agriculture and Technology (IUBAT), Dhaka, Bangladesh

° These authors contributed equally to this work.
* rehana.parvin@iubat.edu

## Abstract

This study delves into the determinants and socioeconomic disparities influencing the utilization of Skilled Birth Attendants (SBAs) for childbirth in Bangladesh, deploying advanced statistical and machine learning (ML) methods. A Multilevel Binary Logistic Regression (MBLR) accounting for hierarchical data structures identified wealth, education levels, and adequate Antenatal Care (ANC) from qualified practitioners as meaningful predictors, while also revealing regional variances—with Khulna and Dhaka divisions and urban areas associated with higher SBA prevalence in contrast to rural areas and divisions like Mymensingh and Sylhet. Socioeconomic inequalities were evaluated using the Wagstaff-type decomposition method. The Concentration Index (CI) and Concentration Curve (CC) indicated that SBA usage was disproportionately higher among affluent households, with ANC visits, wealth index and, schooling contributing most to observed inequity. For prediction purposes, ML algorithms including Artificial Neural Network (ANN), Light Gradient Boosting Machine (LGBM), CatBoost (CATB), and Logistic Regression (LR) and 6 more were trained and gauged utilizing accuracy, precision, recall, F1 score, specificity, and, Area Under the Curve (AUC). ANN achieved the highest AUC (0.81). Model interpretability was enhanced through SHAP (SHapley Additive explanations) plots. The integration of MBLR, Wagstaff-type decomposition and, interpretable ML offers a comprehensive framework for comprehending and anticipating SBA utilization. Findings emphasize the necessity to address socioeconomic imbalances and ensure equitable access to quality ANC, providing actionable evidence for targeted maternal healthcare policies in Bangladesh.

**Data availability statement:** This study is based on publicly available secondary data from the Demographic and Health Surveys (DHS) Program, specifically the 2022 Bangladesh Demographic and Health Survey (BDHS) 2022. The dataset, including the pregnancy recode and geographic data, can be accessed upon request through the DHS Program website: https://dhsprogram.com/data/dataset/Bangladesh_Standard-DHS_2022.cfm?flag=1.

**Funding:** The author(s) received no specific funding for this work.

**Competing interests:** "We hereby confirm that the manuscript has no any actual or potential conflict of interest with any parties, including any financial, personal, or other relationships with other people or organizations within three years of beginning the submitted work that could inappropriately influence or be perceived to influence. We confirm that the paper has not been published previously, it is not under consideration for publication elsewhere, and the manuscript is not being simultaneously submitted elsewhere.".

## Introduction

Maternal mortality remains a significant problem in low- and middle-income countries (LMICs), with poor access to quality health care services [1]. Despite substantial global progress resulting in a 38% reduction in the Maternal Mortality Ratio (MMR) since 2000 [2], women continue to die from mostly preventable causes, including pregnancy and childbirth complications. One of the World Health Organization (WHO)'s sustainable development goals, to be accomplished by 2030, is to achieve an MMR under 70 maternal deaths per 100,000 live births globally [3,4], but countries with weak health systems are still confronted by extremely high numbers of maternal deaths. Limited human and other resources, geographical barriers and, war-induced as well as environmental disasters related displacements inhibit access to life-saving maternal care [1,2,5]. Neonatal outcomes are even worse in those contexts, with 98% of stillbirths and, neonatal deaths projected to occur in LMICs [5–7]. Such a grave reality underscores the need for timely interventions, improved access to Skilled Birth Attendants (SBAs) and other essentials healthcare provisions. Countries have different scopes of midwifery practice, generally defined by national health systems, state or federal government frameworks, professional bodies, and work settings [8].

The United Nations' Sustainable Development Goal (SDG) 3.1 aims to reduce global maternal mortality, with the proportion of births attended by SBAs being one of the key indicators [6,9,10]. In regions such as South and Southeast Asia, where maternal mortality remains alarmingly high compared to other parts of the world despite improvements, investigating the socioeconomic and demographic factors influencing SBA utilization is crucial. In Bangladesh, urban areas have a much higher rate of SBA service delivery compared to rural areas. This study seeks to investigate these determinants using Machine Learning (ML) models and decomposition analysis, to identify the key factors affecting SBA utilization during childbirth in Bangladesh. The research aims to provide actionable insights for improving maternal and neonatal health outcomes by highlighting potential levers for change, including maternal education, wealth, geography, and access to health facilities. To meet global health targets and align with the WHO's objective of reducing the MMR by 2030 [7], it is essential to understand and address these disparities.

Maternal healthcare is a critical indicator of a nation's development, as ensuring safe childbirth is essential for the well-being of both mothers and their newborns [11]. SBA services play a key role in preventing maternal and neonatal mortality by ensuring expert care during childbirth [11,12]. The consistent and effective use of SBAs before, during and, after delivery has proven to be an effective strategy for the timely management of pregnancy problems, the reduction of pregnancy and childbirth related fatalities including, maternal mortality, and the achievement of ideal pregnancy outcomes [13–15]. According to the WHO, globally, maternal mortality is a serious health concern because 287000 cases of women's death are recorded due to pregnancy related issues [16]. About 260000 women's deaths were recorded in 2023 during pregnancy or childbirth, and 92% of cases were from low and LMICs [17]. In 2004, the WHO emphasized that SBA

significantly reduces complications and fatalities associated with childbirth [18]. The primary reason for maternal mortality in LMICs and Bangladesh is the lack of skilled care and SBAs during childbirth [19–21]. Despite improvements in maternal healthcare, Bangladesh continues to face challenges related to maternal mortality, primarily due to limited access to skilled care, particularly in rural and impoverished regions [22,23]. While the country has made substantial progress in increasing SBA usage, significant disparities persist, especially across geographic and socio-economic lines [24].

The government has made a colossal effort to expand the SBA coverage through professional training and programs in rural areas of Bangladesh [12,25]. National statistics indicate the increasing use of SBA [26]; nevertheless, the advantages of SBA are not evenly distributed throughout the country [27]. Urban areas (for example, Dhaka and, Khulna), which have better healthcare facilities and skilled human resources, use more SBA [24]. Relative to rural areas and lower-income regions that are still below sustainable levels of SBA employment [28]. This discrepancy exhibits several underlying features, including restricted access to healthcare facilities, economic limitations and, geographical isolation [29,30]. One of the best ways to lower maternal and infant death rates is to ensure SBA during delivery is used appropriately. Bangladesh is among the first South Asian countries to sign international declarations to improve maternal healthcare, yet it has higher maternal death rates [31]. The ability of women to access SBA services is largely influenced by socioeconomic factors such as income, education levels and, access to health services [32,33].

Notably, underserved rural areas with limited healthcare facilities and transportation options lag behind urban centers in employing SBAs. While tremendous investments have aimed to ameliorate SBA access in these remote areas, the allocation of medical resources remains uneven. Regions with superior infrastructure and greater socioeconomic development, such as Dhaka, demonstrate higher rates of deliveries aided by skilled practitioners, while rural and financially disadvantaged locations persistently face formidable barriers. These disparities in SBA usage underscore the necessity for additional inquiry to pinpoint and address the underlying determinants contributing to these inequities. The findings of this study are critically important for comprehending the factors influencing the utilization of SBAs during childbirth in Bangladesh, specifically pertaining to ongoing imbalances in healthcare access. By distinguishing the socioeconomic, demographic, and medical-related determinants, the study furnishes valuable insights for policymakers striving to improve maternal health outcomes. These understandings are pivotal for formulating policies that advocate equal admission to competent maternal care, irrespective of socioeconomic or regional divergences.

In this research, for the first time to our knowledge, ten widely used ML methods are applied to a dataset involving the use of various ML algorithms for predicting which women are more likely to utilize SBAs. ML provides a data-driven approach to identify high-risk groups and offer tailored solutions, thus optimizing the delivery of maternal health care services. In light of the significant socioeconomic, demographic and health-service related determinants identified in this study, it presents important insights for policymakers involved in improving maternal health outcomes. Consequently, this information is crucial for preparing programs to facilitate universal skilled maternal attendance irrespective of socioeconomic or regional diversities. A secondary purpose is to accentuate the importance of Antenatal Care (ANC) visits by qualified providers towards reaching SBA services. This study focused on the association between sufficient ANC visits and SBA utilization at birth among women. We combine descriptive and statistical models, modern ML techniques with SHAP (Shapley Additive Explanations) based interpretability, and identification of geographic hotspots with decomposition analysis and concentration curve (CC) to advance the current knowledge and offer more accurate predictions along targeted interventions using the latest round of Demographic and Health Survey (DHS) data in Bangladesh. These approaches are crucial for identifying the factors with the greatest impact on SBA utilization and can offer insights into socio-demographic disparities that constrain access to SBAs. The study findings have paramount importance in the maternal health literature, particularly in resource-poor settings like Bangladesh, and will help inform area- and national-level healthcare access policies and programs for reducing morbidity-mortality of mothers as well as for placing half of the sky—universal healthcare coverage to our women.

## Methodology

This study utilizes a mixed-methods approach to investigate the socioeconomic, demographic, and healthcare-related factors influencing the use of SBA during childbirth in Bangladesh. By combining statistical methods and ML techniques, the analysis offers a comprehensive understanding of the determinants of SBA utilization and their implications for maternal healthcare policy.

### Data description

The data were drawn from the Bangladesh Demographic and Health Survey (BDHS) 2022, which was designed by the Bangladesh Bureau of Statistics (BBS) using an Integrated Multi-Purpose Sampling Master Sample. A two-stage stratified sampling design was employed. In the first stage, 675 enumeration areas (EAs) were selected—273 from urban and 438 from rural areas. One rural EA at Cox's Bazar was excluded due to fieldwork challenges. In the second stage, 45 households were systematically selected from each EA, resulting in a sample of 30,330 households. Ever-married women aged 15–49 was interviewed using the women's Questionnaire [26]. The BDHS team collected data in four phases of this nationwide survey, which was carried out from 27 June, 2022–12 December, 2022.

For this study, women aged 15–49 who had experienced at least one childbirth in the 36 months preceding the survey were included. Among these, 17,319 women reported childbirth, with 5,387 providing complete information on delivery attendance. Of these, 597 women who received both SBA and unskilled birth attendant services were excluded due to the complexity of interpreting dual service usage. Additionally, 301 women with missing or outlier data were excluded—250 lacked ANC data and 39 had missing socio-demographic information such as husband's education. The final analytical sample included 4,490 women.

### Study variables

The primary outcome variable is the utilization of SBA during childbirth, coded as a binary variable (1 = Yes, 0 = No). Based on a review of prior research [21,22,24,28,34], relevant explanatory variables were selected for the study. Explanatory variables were grouped into four categories. Economic variables include wealth index (poor, middle, rich), husband's occupation, and woman's occupation (housewife vs others). Socio-demographic variables include woman's age group, age at first birth (before vs after 18 years), education levels of women and their husbands, religion (Islam vs others), household size, relationship to household head (wife vs others), sex of household head, and number of ever born children. Geographic variables include division, district, and place of residence (urban vs rural). Healthcare-related variables include adequacy of ANC visits, toilet facility type, history of pregnancy loss, and media access.

### Data preprocessing

Missing data were handled by exclusion, especially for variables with significant missingness (e.g., ANC visits). Variables were transformed as needed—for instance, the wealth index was recategorized into three groups. The Generalized Variance Inflation Factor (GVIF) was used to check multicollinearity; all values were below 2, suggesting no collinearity issues [35]. For ML modeling, data were split into 80% training and 20% testing sets. After splitting, the training set was imbalanced (1,087 "No" vs 2,057 "Yes" cases). The Synthetic Minority Over-sampling Technique (SMOTE) is used to balance class distribution (2,334 "No" vs 2,351 "Yes" cases) only on training dataset [36]. The overview of research flow and ML framework is presented in Fig 1. All analyses were performed using software R (version 4.5.1) and STATA (version 17).

### Statistical Analysis

Descriptive statistics were computed to summarize key characteristics of the study population, and spatial distributions were visualized in the map of Bangladesh. Bivariate analysis was conducted using chi-square tests to identify

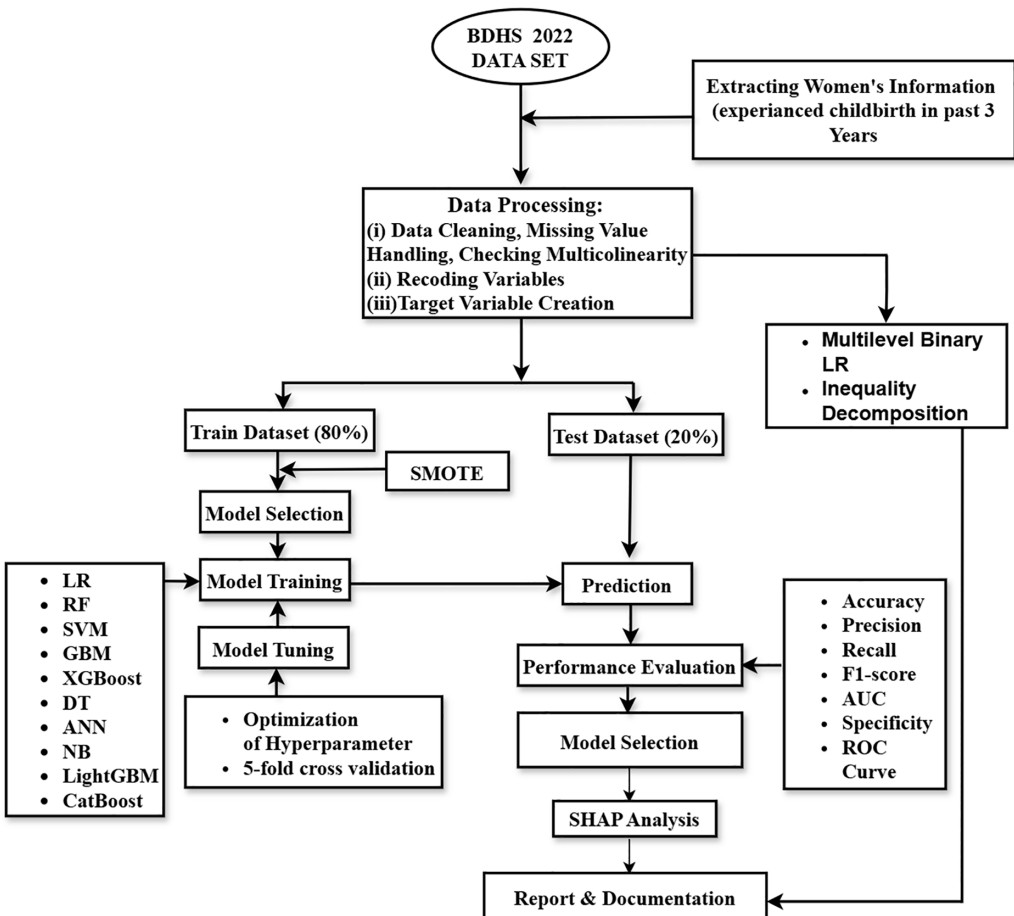

**Fig 1. Overview of research flow and ML framework.**

associations between the outcome and explanatory variables [37]. Given the hierarchical structure of the data, with women nested within clusters, multilevel binary logistic regression (MBLR) analysis was performed using the "melogit" command in STATA. The intraclass correlation coefficient (ICC) was calculated as 0.1207, exceeding the 0.10 threshold for multilevel modeling [22,38,39]. A likelihood ratio test further supported the use of MBLR over a standard logistic regression (LR) model (Chi-Square = 54.08; $p < 0.001$) [40].

## Inequality measurement

To assess inequality in SBA utilization, decomposition analysis was conducted using a Wagstaff-type decomposition method by using "margins" and "ineq" libraries in R for measuring the Concentration Index (CI), constructing the CC and decomposing inequality. Since the outcome variable is binary, quasibinomial LR with survey weights was applied to account for clustering [41,42]. CC and CI were employed to measure inequality across socioeconomic groups [41,43]. The Percentage Contribution (PC) of each covariate was obtained by dividing its absolute contribution in the Wagstaff decomposition by the total explained concentration index and expressing the result as a percentage. The details about the decomposition setup (pseudo code) listed in the supporting information (S1 Table).

## Machine learning models

In addition to traditional statistical methods, 5-fold cross-validation with random search for tuning hyperparameters (tune length of 10) was used to train 10 ML models with appropriate train control using the train dataset and model performance was justified by evaluating the confusion matrix using the test dataset. These models are used to predict the likelihood chance and to find out influential determinants of SBA usage. LR is a supervised classification technique that estimates probabilities for binary outcomes and uses a decision threshold of 0.5 to classify observations. The LR model is one of the most popular and reliable statistical models due to its high efficiency and interpretive nature [44]. Support Vector Machine (SVM) is one of the most widely used classifiers because it can easily handle complex and nonlinear data as well as basic multi-class problems; it is able to map the data into higher dimensionalities [45]. While Decision Tree (DT) is intuitive model that recursively split data into partitions based on the values of each feature [46,47], the Random Forest (RF) is an ensemble approach that creates multiple decision trees to reduce overfitting and encourage diversity. CatBoost (CATB) is also a type of ML method called Gradient Boosted DT. It has been useful in studies of big data. It works well with categorical data because it is based on a DT. It has been utilized for both classification and regression tasks across various contexts [48]. Naïve Bayes (NB) is a simple probabilistic classifier based on the assumption of independence among predictors and is highly scalable to large datasets, making it very useful for applications such as text classification [49]. Artificial Neural Networks (ANN) − ANNs are flexible brain-inspired models that can learn complex relationships. In this study, a feed-forward multilayer perceptron architecture, with model parameters refined through cross-validation was employed. ANNs are very powerful, but they can also learn too much if they aren't properly regularized. [50]. The Gradient Boosting Machine (GBM) creates a strong predictor by boosting multiple weak learners in sequence and optimizing loss by gradient descent and works exceptionally on complex non-linear tasks [51]. Notably, extreme gradient boosting (XGBoost), is an efficient implementation of GBM with support for parallel computation, regularization, and methods that allow handling large datasets [52]. Each of these models, with its advantages, is used in different domains to increase accuracy and manage data complexities. LightGBM (LGBM) is a fast version of Gradient Boosting Decision Trees that uses Gradient-based One-Side Sampling and Exclusive Feature Bundling to focus on important events, reducing the number of features. The model is made up of a series of decision trees that are built one after the other with hyperparameters like the number of leaves, tree depth, learning rate, and minimum samples per leaf [53]. All R libraries used for model training are listed in the supporting information (S2 Table).

## Evaluation metrics

Model evaluation is carried out using a variety of performance metrics calculated from the confusion matrix of all models, such as accuracy, precision, recall, F1 score, Receiver Operating Characteristic (ROC) curve, Area Under the Curve (AUC), and specificity. Accuracy measures the ratio of correctly classified instances, while precision measures the number of predicted positive cases. Recall captures the proportion of true positive cases identified. The F1 score balances precision and recall, while AUC assesses the model's overall ability to distinguish between classes by calculating the area beneath the ROC curve—a graphical plot of true positive rate against false positive rate. Specificity measures the model's accuracy in identifying true negative cases [54]. We considered the AUC value for justifying models' performance. In cases of equal AUC value, we considered the accuracy metric for model justifications. The DeLong's test is used for comparing statistical differences in AUC values among models [55].

## Model Interpretability

KernelSHAP, a model-agnostic kernel-based method used in this study. The SHAP analysis was applied by Lundberg and Lee to enhance the model's interpretability by characterizing every feature's contribution to model prediction. SHAP values are used to create a single value explaining the impact of each feature on the model's output, especially in non-linear ML

models [56]. SHAP values were calculated for different models by using the library "kernelSHAP" in R and SHAP feature importance (Bar) plot constructed to understand the most influential determinants of SBA. Also, SHAP waterfall plot, Beeswarm and Dependency for finding out key determinants which were responsible for SBA predictions, for demonstrating non-linear relationships and individual instances. These plots were created using the library "shapviz" in R and they explained how the determinants contribute to the model's output. The details and pseudo code of the SHAP implementation are provided in the supporting information (S1 Table).

### Ethics Statement

The BDHS was implemented by the National Institute of Population Research and Training (NIPORT) of the Ministry of Health and Family Welfare, with technical assistance from ICF International, and received ethical approval from the Institutional Review Board of ICF International and the National Research Ethics Committee of the Bangladesh Medical Research Council (BMRC). Written informed consent was obtained from all participants during the original survey.

## Result and discussion

This study aims to investigate the factors influencing the utilization of SBA during childbirth in Bangladesh, with a particular focus on regional and socioeconomic disparities. The findings highlight the significant geographic variability in SBA-assisted childbirth, underscoring the challenges faced by different divisions in ensuring equitable access to skilled maternal care. The study incorporates both statistical analyses and geographical visualizations to provide a comprehensive overview of SBA prevalence across the country. By examining various determinants such as wealth index, education, antenatal care, and geographic location etc., the study aims to identify the barriers preventing widespread access to SBA for the people all over the country.

### Results of statistical analysis

Fig 2 offers a concise visualization of the geographical disparities and inconsistencies in the rate of SBAs during childbirth across Bangladesh. In Fig 2 the SBA coverage by division map elucidates substantial regional variation in SBA rates, with areas exhibiting remarkably high and remarkably low levels of SBA. The insights from the map corroborate earlier statistical examinations, emphasizing the disparities in SBA utilization between diverse divisions. Take, for instance, zones like Khulna, which stand out with a higher percentage of 84% of SBA-guided births followed by Dhaka and Rajshahi divisions with 72% and 71% decent percentages of SBA-guided births. But in other divisions, the goal of the 4th Health, Population, and Nutrition Sector Program (HPNSP)'s (65% rate of SBA-assisted childbirth by 2023) was not achieved, the red zone of the map demonstrated significantly low SBA usage in these regions, pointing to barriers that hinder access to SBA. These could involve issues like constrained healthcare amenities, economic restrictions, or geographic isolation, which contribute to the underutilization of SBA services. Remarkably low prevalence was found in the Mymensingh (54%) and Sylhet (54.8%) divisions.

The prevalence of SBA by district map differentiated districts through color coding, showing that locales marked in green generally benefit from enhanced access to medical care (SBA rate over 70%). In stark contrast, regions denoted in red, signifying under 60% SBA prevalence, highlight critical shortcomings in healthcare provision. Maroon dots indicate SBA rate of 60–64%, just under the goal of 4th HPNSP and yellow indicates the areas just over the 65% of SBA rate. These districts exhibited a robust healthcare framework and greater access to trained experts, contributing to their elevated SBA rates. Almost every district of Khulna had green dots except one, which was much closer to the goal. On the other hand, almost every region of Chattogram's hill track areas, Mymensingh and Sylhet divisions, most of the regions of Rangpur, Barishal and other parts of Chattogram division and a few regions of Rajshahi and Dhaka divisions were showing red and maroon dots. The 4th HPNSP goal was not achieved in more than 40% of districts in Bangladesh. About 10% of the district's prevalence is slightly over the target.

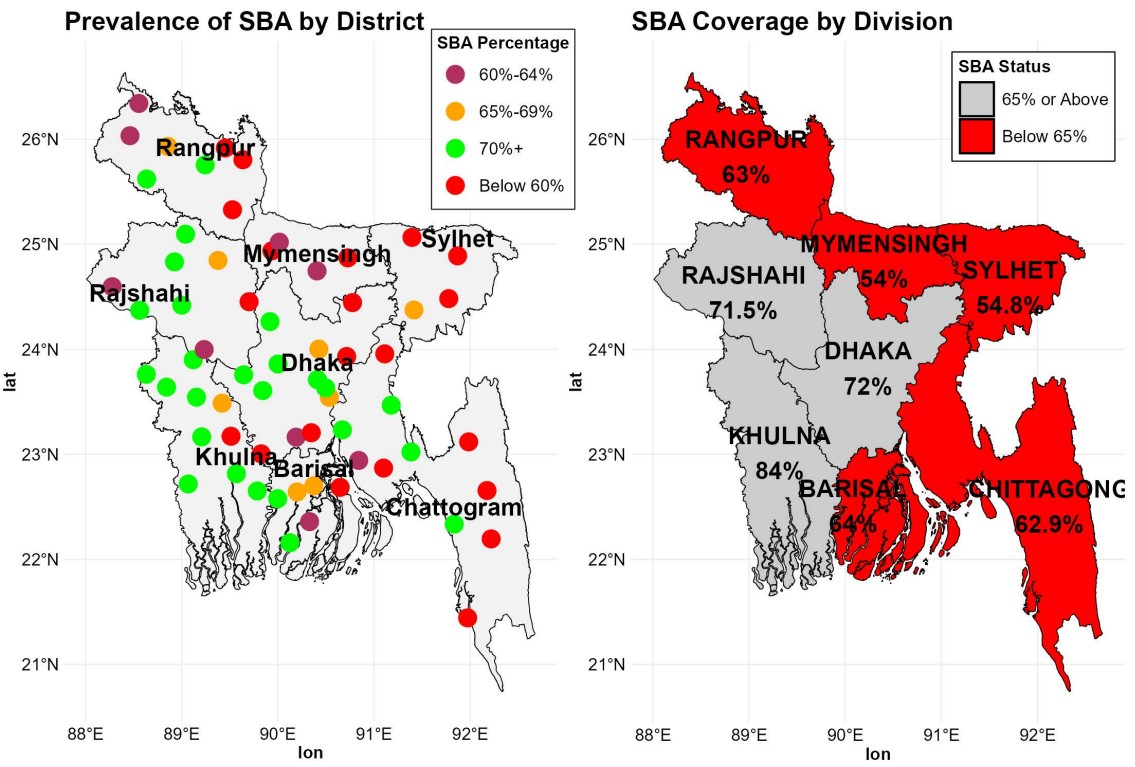

**Fig 2. Prevalence of SBA based on demographic areas in Bangladesh.**

The results outlined in Table 1 offer a comprehensive overview of the variables influencing SBA utilization during childbirth in Bangladesh. This table highlights the distribution of SBA-assisted deliveries across different demographic and socioeconomic indicators, providing meaningful insights into existing disparities and linkages.

Place of residence emerged as a pivotal determinant, with a marked difference between urban and rural areas. The highest proportion of women with SBA usage was from the Dhaka division (16.47%), followed by Chattogram (16.07%), Khulna (13.72%), Rangpur (11.23%), Rajshahi (11.10%), Barishal (10.89%), Mymensingh (10.89%) and Sylhet (9.63%). This finding underscores the geographic barriers to the healthcare facilities during childbirth and significantly impacts SBA access (Chi-square = 156.527; $p \leq 0.001$). Most of the women assisted by an SBA during childbirth were from rural areas 67.39% and 32.61% from urban areas, which had an effect on SBA usage during childbirth ($p \leq 0.001$). The wealth index additionally proved to be a key aspect, with notable ties between economic standing and SBA utilization. Women who experienced SBA-assisted birth mostly had a higher wealth index of 49.39% compared to 30.60% among the poor, highlighting the role financial resources play in enabling access to an SBA-assisted birth (Chi-square = 490.8639; $p \leq 0.001$).

Education played a crucial role in access to services from SBA for both women and their husbands in Bangladesh. The analysis revealed that higher levels of education among women correlated with a greater likelihood of assistance from SBAs during childbirth. 79.81% of women had at least a secondary level of education among the women who had experienced SBA-assisted birth. Husbands' education played a part too, as couples with more educated husbands also tended to utilize SBAs more. The enormous chi-square values of 440.6119 ($p \leq 0.001$) for women's education and 452.3079 ($p \leq 0.001$) for their partners' schooling emphasize how important education is to maternal healthcare choices in remote areas.

**Table 1. Bivariate distribution of SBA across all independent variables.**

| Study Variables | No (%) | Yes (%) | Overall (%) | Chi-Square value |
|---|---|---|---|---|
| | 1552(34.57%) | 2938(65.43%) | 4490(100%) | |
| **Living location** | | | | 156.527*** |
| Barishal | 180(11.60) | 320(10.89) | 500(11.14) | |
| Chattogram | 278(17.91) | 472(16.07) | 750(16.70) | |
| Dhaka | 188(12.11) | 484(16.47) | 672(14.97) | |
| Khulna | 77(4.96) | 403(13.72) | 480(10.69) | |
| Mymensingh | 272(17.53) | 320(10.89) | 592(13.18) | |
| Rajshahi | 130(8.38) | 326(11.10) | 456(10.16) | |
| Rangpur | 194(12.50) | 330(11.23) | 524(11.67) | |
| Sylhet | 233(15.01) | 283(9.63) | 516(11.49) | |
| **Place of residence** | | | | 122.0601*** |
| Urban | 341(21.97) | 1,123(38.22) | 1,464(32.61) | |
| Rural | 1,211(78.03) | 1,815(61.78) | 3,026(67.39) | |
| **Wealth index** | | | | 490.8639*** |
| Poor | 967(62.31) | 899(30.60) | 1,866(41.56) | |
| Middle | 293(18.88) | 588(20.01) | 881(19.62) | |
| Rich | 292(18.81) | 1,451(49.39) | 1,743 (38.82) | |
| **Women's education level** | | | | 440.6119*** |
| No education | 147(9.47) | 92(3.13) | 239(5.32) | |
| Primary | 553(35.63) | 501(17.05) | 1,054(23.47) | |
| Secondary | 761(49.03) | 1,596(54.32) | 2,357(52.49) | |
| College or Higher | 91(5.86) | 749(25.49) | 840(18.71) | |
| **Husband's education level** | | | | 452.3079*** |
| No education | 374(24.10) | 314(10.69) | 688(15.32) | |
| Primary | 630(40.59) | 707(24.06) | 1,337(29.78) | |
| Secondary | 441(28.41) | 1,100(37.44) | 1,541(34.32) | |
| College/Higher | 107(6.89) | 817(27.81) | 924(20.58) | |
| **Toilet facility** | | | | 69.6475*** |
| Unhygienic | 564(36.34) | 720(24.51) | 1,284(28.60) | |
| Hygienic | 988(63.66) | 2,218(75.49) | 3,206(71.40) | |
| **Religion** | | | | 55.4091*** |
| Muslim | 1,484(95.62) | 2,616(89.04) | 4,100(91.31) | |
| Non-Muslim | 68(4.38) | 322(10.96) | 390(8.69) | |
| **Relation with household head** | | | | 50.8851*** |
| Wife | 958(61.73) | 1,486(50.58) | 2,444(54.43) | |
| Others | 594(38.27) | 1,452(49.42) | 2,046(45.57) | |
| **Sex of household head** | | | | 0.4189 |
| Male | 1,386(89.30) | 2,605(88.67) | 3,991(88.89) | |
| Female | 166(10.70) | 333(11.33) | 499(11.11) | |
| **Husband's occupation** | | | | 153.3149*** |
| Unemployed | 23(1.48) | 71(2.42) | 94(2.09) | |
| Farmer | 437(28.16) | 410(13.96) | 847(18.86) | |
| Hard worker | 507(32.67) | 975(33.19) | 1,482(33.01) | |
| Service | 354(22.81) | 816(27.77) | 1,170(26.06) | |
| Business | 231(14.88) | 666(22.67) | 897(19.98) | |

*(Continued)*

**Table 1.** (Continued)

| Study Variables | No (%) | Yes (%) | Overall (%) | Chi-Square value |
|---|---|---|---|---|
| | 1552(34.57%) | 2938(65.43%) | 4490(100%) | |
| **Women's occupation** | | | | 50.8963*** |
| Housewife | 1,059(68.23) | 2,291(77.98) | 3,350(74.61) | |
| Others | 493(31.77) | 647(22.02) | 1,140(25.39) | |
| **Media access** | | | | 346.0805*** |
| No access | 871(56.12) | 818(27.84) | 1,689(37.62) | |
| Have access | 681(43.88) | 2,120 (72.16) | 2,801(62.38) | |
| **History of pregnancy losses** | | | | 7.273** |
| No records | 1,272(81.96) | 2,308(78.56) | 3,580(79.73) | |
| Have records | 280(18.04) | 630(21.44) | 910(20.27) | |
| **Total ever born children** | | | | 239.1917*** |
| 1 child | 400(25.77) | 1,245(42.38) | 1,645(36.64) | |
| 2 children | 500(32.22) | 1,074(36.56) | 1,574(35.06) | |
| 3+children | 652(42.01) | 619(21.07) | 1,271(28.31) | |
| **Household size** | | | | 6.7244** |
| ≤4 members | 462(29.77) | 982(33.42) | 1,444(32.16) | |
| 5-6 members | 639(41.17) | 1,121(38.16) | 1,760(39.20) | |
| more than 7 members | 451(29.06) | 835(28.42) | 1,286(28.64) | |
| **ANC visit** | | | | 372.8225*** |
| Inadequate | 1,225(78.93) | 1,445(49.18) | 2,670(59.47) | |
| Adequate | 327(21.07) | 1,493(50.82) | 1,820(40.53) | |
| **Age at first birth** | | | | 58.2544*** |
| <18 years | 601(38.72) | 811(27.60) | 1,412(31.45) | |
| 18+years | 951(61.28) | 2,127(72.40) | 3,078(68.55) | |
| **Women's age group** | | | | 10.6003** |
| <20 years | 198(12.76) | 369(12.56) | 567(12.63) | |
| 20-29 | 913(58.83) | 1,837(62.53) | 2,750(61.25) | |
| 30-39 | 407(26.22) | 695(23.66) | 1,102(24.54) | |
| 40-49 | 34(2.19) | 37(1.26) | 71(1.58) | |

***for $p$-value ≤ 0.001, **for $p$-value ≤ 0.05

The type of toilet available is also strongly associated with SBA usage, three-quarters of women with access to sanitary facilities in their homes used SBAs for birth compared to less than a quarter of those lacking proper plumbing. Clearly, the surrounding environmental conditions, such as standards of hygiene, had an impact on maternal health outcomes ($p \leq 0.001$). Though sex of household head was not found significant ($p = 0.517$), relation with household head and religion substantially impacted rates of SBA utilization. Cultural norms or uneven healthcare access within religious communities likely account for some disparity in maternal care decisions. The chi-square test suggested that religion and relation with household head wielded modest influence on SBA use, but further probing into cultural traditions and medical access remains necessary.

Both women's work status and their husbands' occupations markedly shaped reliance on SBAs ($p \leq 0.001$). Housewives most commonly received skilled help at birth (77.98%), perhaps owing to heavier dependency on services and more free time to seek treatment. Conversely, women employed elsewhere saw lower SBA usage (22.24%). Mirroring this, husbands engaged in business (22.67%) tended to have wives aided by capable attendants, compared to jobless spouses

(2.42%). Access to mass media also significantly influenced SBA usage, 72.16% of women had access to media, e.g., smartphone, television and internet among those women who had experienced SBA-assisted birth ($p \leq 0.001$). Exposure to information undoubtedly bolsters awareness of skilled care's advantages during delivery, equipping women with vital knowledge of available options. Also, having a past history of pregnancy losses was associated with the usage of SBA during childbirth ($p = 0.007$). The percentage of women with 1 child (42.38%) is double that of other women who had 3 or more children (21.07%) with a significant $p$-value of 0.001. Women who experienced SBA-assisted birth, a higher proportion of them were between 20 and 29 years (62.53%). The factor along with factors household size and woman's age at first birth also showed a significant impact on SBA usage during childbirth. Finally, proper ANC proved one of the strongest correlates of SBA use (chi-square = 427.134; $p \leq 0.001$)). This indicates that quality prenatal monitoring strongly correlates with skilled delivery access, better informing and preparing women.

Table 2 presents an analysis of the socioeconomic and demographic factors that influence SBA-assisted childbirth in Bangladesh, with a focus on Adjusted Odd Ratio (AOR), Marginal Effect (ME), and their significance. The results shed light on the key drivers of SBA utilization, reflecting the significant disparities in access to maternal healthcare across different population groups. These findings provide a clear understanding of how various factors influence the availability and use of SBAs and contribute to maternal health outcomes in Bangladesh.

The differences in maternal care access among divisions were huge, as indicated in table 2. Khulna was one of the regions with high odds of access to SBAs (AOR = 4.06) i.e., people in Khulna were 4.06 times more likely to access SBA compared to Mymensingh. Such discrepancies likely relate to the varying capabilities of national health systems, skills shortages, and regional health policies. Similarly, Rajshahi (AOR = 1.89) and Dhaka (AOR = 1.47) were also more likely to use SBAs as compared to Mymensingh division. The above evidence emphasizes attaining regional equity through high and equal access to maternal care services, with more availability and better quality of maternal healthcare especially for the poor of rural divisions, as urban show a 1.25 times higher likelihood than rural areas, suggesting an important role of urbanization and healthcare infrastructure in urban areas in delivering access to skilled care.

Wealth index played a notable role in how SBAs were utilized. Women from rich household (AOR = 1.95) and those in the middle class (AOR = 1.33) were more inclined to make use of SBAs compared to their counterparts in impoverished circumstances. This aligns with the broader socioeconomic architecture in Bangladesh, where more affluent individuals have better means to pay for private health services, including skilled obstetric care. In stark contrast, lower-income households, particularly those residing in rural locales, confront obstacles in accessing quality maternal healthcare, generally due to the restricted availability of skilled practitioners in public health settings. The substantial wealth disparities underscored by these odds ratios highlight the necessity for policies that improve SBA access for populations with meager incomes and rural inhabitants, ensuring that financial circumstances are not an impediment to quality maternal healthcare.

The role of education was equally critical in determining access to SBAs. The odds of accessing SBAs increased dramatically with higher levels of education. Women whose husbands hold a college/higher level of education were 1.8 times more likely to have an SBA-assisted delivery than women whose husbands had no education. Women with college/higher and secondary education were 2.71 times and 1.5 times more likely to utilize SBAs compared to those with no education. This finding underscores the impact of education on health-seeking behavior. Educated women tend to have better health literacy, enabling them to understand the importance of SBAs and seek professional care during pregnancy and childbirth. In Bangladesh, where female education remains a significant challenge, especially in rural areas, improving access to education for women is vital to enhancing maternal health outcomes. Educating women about the benefits of SBA and other maternal health services is a key intervention strategy to improve maternal healthcare utilization.

Not only education but also occupation played a crucial role. Housewife women had 1.54 times more odds of an SBA-assisted delivery than those women with other professions (AOR = 1.54). Also, their husbands who were hard workers were 1.26 times (AOR = 1.26) and businessmen were 1.43 times (AOR = 1.43) more likely to have an SBA-assisted childbirth than those women whose husbands were farmers. A housewife or non-working woman is very much focused

**Table 2. Multilevel binary logistic regression analysis results.**

| Study variables | Categories | AOR (95% CI) | ME (95%CI) | PC |
|---|---|---|---|---|
| **Living location** | Barishal | 1.46(0.99,2.13) | 0.062(−0.001,0.125) | 13.26 |
| | Chattogram | 1.17(0.82,1.66) | 0.026(−0.034,0.085) | |
| | Dhaka** | 1.47(1.02,2.11) | 0.064(0.004,0.124) | |
| | Khulna*** | 4.06(2.68,6.17) | 0.209(0.149,0.268) | |
| | Mymensingh | **Reference** | | |
| | Rajshahi*** | 1.89(1.28,2.78) | 0.103(0.04,0.165) | |
| | Rangpur | 1.42(0.98,2.06) | 0.058(−0.003,0.12) | |
| | Sylhet | 0.99(0.68,1.44) | −0.002(−0.067,0.06) | |
| **Place of residence** | Urban** | 1.25(1,1.56) | 0.035(0,0.07) | 0.88 |
| | Rural | **Reference** | | |
| **Wealth index** | Poor | **Reference** | | 7.56 |
| | Middle** | 1.33(1.08,1.64) | 0.048(0.012,0.084) | |
| | Rich*** | 1.95(1.55,2.44) | 0.108(0.071,0.146) | |
| **Women's education level** | No education | **Reference** | | 5.87 |
| | Primary | 1.19(0.83,1.7) | 0.029(−0.033,0.091) | |
| | Secondary** | 1.5(1.05,2.15) | 0.069(0.006,0.131) | |
| | College/Higher*** | 2.71(1.71,4.28) | 0.159(0.084,0.234) | |
| **Husband's education level** | No education | **Reference** | | 4.22 |
| | Primary | 0.95(0.76,1.2) | −0.008(−0.046,0.031) | |
| | Secondary | 1.23(0.97,1.58) | 0.034(−0.006,0.075) | |
| | College/Higher*** | 1.8(1.28,2.54) | 0.093(0.039,0.147) | |
| **Toilet facility** | Unhygienic | **Reference** | | 1.24 |
| | Hygienic** | 1.24(1.03,1.48) | 0.034(0.005,0.063) | |
| **Religion** | Muslim | **Reference** | | 8.59 |
| | Non-Muslim*** | 3.04(2.13,4.33) | 0.158(0.115,0.202) | |
| **Household size** | ≤4 members | 1.07(0.85,1.35) | 0.01(−0.026,0.047) | 0.08 |
| | 5-6 members | 1.05(0.86,1.28) | 0.007(−0.024,0.039) | |
| | ≥7 members | **Reference** | | |
| **Relation with household head** | Wife | **Reference** | | 0.8 |
| | Others | 1.2(0.99,1.46) | 0.029(−0.001,0.06) | |
| **Media access** | No access | **Reference** | | 3.94 |
| | Have access*** | 1.44(1.21,1.71) | 0.059(0.03,0.087) | |
| **Total ever born children** | 1 child*** | 3.06(2.31,4.05) | 0.182(0.137,0.228) | 13.92 |
| | 2 children*** | 1.83(1.46,2.3) | 0.103(0.064,0.142) | |
| | ≥3 children | **Reference** | | |
| **Age at first birth** | <18 years | 1.12(0.93,1.35) | 0.018(−0.011,0.047) | 0.34 |
| | 18 + years | **Reference** | | |
| **History of pregnancy losses** | No records | **Reference** | | 4.06 |
| | Have records*** | 1.54(1.26,1.87) | 0.066(0.036,0.096) | |
| **Women's age group** | <20 years | 0.56(0.27,1.14) | −0.093(−0.203,0.017) | 3.96 |
| | 20-29 years | 0.90(0.47,1.72) | −0.016(−0.115,0.083) | |
| | 30-39 years | 1.20(0.63,2.28) | 0.027(−0.070,0.125) | |
| | 40-49 years | **Reference** | | |

*(Continued)*

**Table 2.** (Continued)

| Study variables | Categories | AOR (95% CI) | ME (95%CI) | PC |
|---|---|---|---|---|
| **Husband's occupation** | Unemployed | 1.81(0.98,3.36) | 0.093(0.001,0.185) | 2.11 |
| | Farmer | **Reference** | | |
| | Hard worker** | 1.26(1,1.58) | 0.037(0,0.074) | |
| | Service | 1.26(0.99,1.61) | 0.038(−0.002,0.077) | |
| | Business** | 1.43(1.1,1.86) | 0.058(0.016,0.099) | |
| **Women's occupation** | Housewife*** | 1.54(1.28,1.84) | 0.069(0.039,0.099) | 4.89 |
| | Others | **Reference** | | |
| **ANC by qualified person** | Inadequate | **Reference** | | 24.31 |
| | Adequate*** | 2.53(2.12,3.02) | 0.149(0.121,0.176) | |
| **Constant*** | | 0.08(0.04,0.16) | | |
| **Cluster variable** | Var(constant) | 0.45(0.31,0.67) | | |

Wald chi2(35) =715.27(P-value ≤ 0.001); AUC=0.8543

AOR= Adjusted Odd Ratio; ME= Marginal Effect; PC= Percentage Contribution

***for *p*-value ≤ 0.001, **for *p*-value ≤ 0.05

on domestic life and being unemployed gave her a lot of flexibility to attend more ANC visits. Husband's occupation, like business ensures a lot of advantages for women, e.g., higher socioeconomic condition, better education, urban residence and positive health attitudes collectively lead to a higher chance of SBA-assisted childbirth.

Adequate ANC by qualified personnel was another major determinant of SBA usage. Women who receive adequate ANC (AOR = 2.53) were significantly more likely to deliver with the assistance of an SBA. This highlights the importance of having skilled professionals not only during delivery but also throughout the pregnancy to monitor and manage potential complications. In Bangladesh, where ANC services are available in both urban and rural areas, ensuring that these services are of high quality and provided by skilled professionals is crucial to increasing SBA utilization. The results emphasized the need to strengthen ANC services across Bangladesh, ensuring that women receive proper prenatal care to prevent complications during delivery. The number of children ever born also played a significant role in determining access to SBAs. Women with one child (AOR = 3.06) and two children (AOR = 1.83) were more likely to access SBAs compared to women with three or more children. Smaller families tend to have more resources available for healthcare, including SBAs, as families with fewer children can allocate more resources per child. This finding underscores the importance of family planning programs in Bangladesh, as smaller family sizes can ease the financial burden and improve access to maternal care.

It was also found that religious differences played an important role in SBA access, where non-Muslim respondents had higher chances to use SBA services (AOR = 3.04). This might be a result of socio-cultural causes or some side of discrimination in the health system that negatively affects the accessibility of maternal health care services to Muslim women in some areas. Thus, the need for inclusive healthcare policies focusing on all religious groups within Bangladesh, to address maternal healthcare coverage, is paramount. Results point to obstacles existing for Muslim communities, such as lack of awareness, obstacles to healthcare access, or cultural customs limiting their use of SBAs.

The absence of unhygienic toilets (AOR = 1.24) supported maternal health through a lower incidence of infections and complications at birth. Access to clean toilets in rural and urban endemic areas of Bangladesh, where sanitation remains a big challenge, is a step in the right direction towards improved maternal and perinatal outcomes. As a result, women could be more likely to develop infections whilst giving birth in these sites with poor sanitation, influencing their willingness to access professional care. That is why improving sanitation in rural and distant areas has become an important part of healthy living. An AOR of 1.44 suggested that access to media has been another important contributor to seeking skilled

maternal care, when and if needed. The media also played a significant role in raising awareness regarding maternal health and the importance of SBAs and access to health services. Broadening media coverage, particularly in rural locations, may increase awareness around the availability of SBAs and alleviate concerns or preconditions to procuring care.

Mothers with a history of reproductive failures (AOR = 1.54) were likely to seek SBAs in later pregnancies. While this study highlighted the importance of SBAs for all births in a population with a high burden of complications, it reinforces the importance of ensuring that women who have a complication history receive care with skilled personnel so that the risk of future maternal and neonatal morbidity remains low. We need to provide these women with the best possible care and support to further reduce pregnancy losses and improve maternal health. This study's results made a whole lot of sense and clearly imply that access to SBAs is critical in addressing maternal health in Bangladesh. Important factors like regional differences, socioeconomic status, literacy, hygiene, and media exposure critically influence the probability of availing SBAs. Reducing maternal and child mortality in Bangladesh requires targeted interventions to address these socioeconomic and demographic inequalities, including greater education, improved healthcare infrastructure, awareness, and family planning services. Civil society organizations working to narrow health access disparities must prioritize access to SBAs for all pregnant women, regardless of their socioeconomic status or location in the country, as this will lead to safer pregnancies and deliveries and ultimately better maternal and child health outcomes in the country.

### Results of decomposition analysis

The analysis presented in Table 3 uses Wagstaff's decomposition of the CI to examine the socioeconomic inequalities in the utilization of SBA during childbirth in Bangladesh. This method breaks down the contribution of various socioeconomic factors, allowing researchers to learn how different determinants impact SBA access and maternal health outcomes. The CI, elasticity, absolute contribution and PC reflect the extent to which each variable contributes to socioeconomic inequalities in SBA utilization. The regional disparities in Bangladesh showed varying impacts on SBA usage across divisions. The combined contribution of Khulna (3.9%) and Dhaka (2.74%) to overall CI is 6.6%, indicating the significant role of these regions in socioeconomic inequality in access to SBA service. Which is a clear reflection of demographical disparity, meaning women who live in these regions were more likely to use SBA and this is concentrated among wealthier women in these regions. A similar type of pattern is also reflected among women live in urban areas (PC = 3.02%).

Wealth also played a crucial role in determining access to SBAs. The rich category showed highest contribution of 30.85% to the inequality, which suggested that wealthier individuals had far greater access to SBA services compared to their poor counterparts, who were often limited to government healthcare systems that might not provide adequate skilled care, especially in rural areas. These findings highlighted the importance of addressing wealth-based disparities in healthcare to improve access to SBAs for lower-income populations in Bangladesh. Education was another significant factor influencing SBA usage by wealthier women. Women with college/higher education showed a 10.24% contribution to CI, which highlighted that educated women were more likely to understand the importance of SBA services and seek professional care during childbirth, which underscored the need to enhance female education in Bangladesh, particularly in rural areas where education levels are still low. Respondent's husband having secondary (PC = 1.83%) and college/higher education (PC = 9.68%) showed higher levels of SBA utilization among their wives. This reflects that educated husbands may play a role in encouraging women to seek professional care during childbirth, further emphasizing the importance of family education.

The type of toilet facility was a contributing factor, with those having access to hygienic toilets (PC = 3.45%) were more likely to utilize SBAs. Access to proper sanitation (moderately concentrated among wealthier women) reduces the risk of infections and other complications during childbirth, leading to better maternal health outcomes. Improving sanitation infrastructure, especially in rural Bangladesh, is crucial to ensuring that women are not only protected from infections but also encouraged to seek SBAs. The number of ever born children was another determinant, with women who had 1 child (PC = 4.25%) showing a good contribution to CI of accessing SBA. Smaller families tend to have more resources available

**Table 3. Wagstaff decomposition of concentration index for measuring socioeconomic inequalities in SBA assisted childbirth in Bangladesh.**

| Study Variable | Categories | CI | Elasticity | Contribution to CI | |
|---|---|---|---|---|---|
| | | | | AC | PC |
| **Living location** | Barishal | −0.4018 | 0.0022 | −0.0009 | −0.6464 |
| | Chattogram | −0.1569 | 0.0027 | −0.0004 | −0.3045 |
| | Dhaka | 0.2235 | 0.0168 | 0.0038 | 2.7358 |
| | Khulna | 0.171 | 0.0313 | 0.0053 | 3.8996 |
| | Mymensingh | **Reference** | | | |
| | Rajshahi | 0.0987 | 0.0159 | 0.0016 | 1.1441 |
| | Rangpur | −0.1003 | 0.0088 | −0.0009 | −0.646 |
| | Sylhet | 0.0984 | −0.0005 | 0 | −0.0354 |
| **Place of residence** | Urban | 0.3128 | 0.0133 | 0.0041 | 3.0237 |
| | Rural | **Reference** | | | |
| **Wealth index** | Poor | **Reference** | | | |
| | Middle | 0.0373 | 0.0171 | 0.0006 | 0.4644 |
| | Rich | 0.6192 | 0.0683 | 0.0423 | 30.8512 |
| **Women's education level** | No education | **Reference** | | | |
| | Primary | −0.2584 | 0.0063 | −0.0016 | −1.1817 |
| | Secondary | 0.0134 | 0.0532 | 0.0007 | 0.5193 |
| | College/Higher | 0.4176 | 0.0336 | 0.014 | 10.2365 |
| **Husband's education level** | No education | **Reference** | | | |
| | Primary | −0.2291 | 0.0002 | −0.0001 | −0.0409 |
| | Secondary | 0.0964 | 0.026 | 0.0025 | 1.8308 |
| | College/Higher | 0.4255 | 0.0312 | 0.0133 | 9.6768 |
| **Toilet facility** | Unhygienic | **Reference** | | | |
| | Hygienic | 0.0982 | 0.0481 | 0.0047 | 3.4458 |
| **Religion** | Muslim | **Reference** | | | |
| | Non-Muslim | −0.0169 | 0.0213 | −0.0004 | −0.2619 |
| **Household size** | ≤4 members | −0.0242 | 0.0117 | −0.0003 | −0.2062 |
| | 5-6 members | −0.0235 | 0.011 | −0.0003 | −0.189 |
| | ≥7 members | **Reference** | | | |
| **Relation with household head** | Wife | **Reference** | | | |
| | Others | 0.0644 | 0.0263 | 0.0017 | 1.2377 |
| **Total ever born children** | 1 child | 0.0583 | 0.1 | 0.0058 | 4.252 |
| | 2 children | 0.035 | 0.0542 | 0.0019 | 1.382 |
| | ≥3 children | **Reference** | | | |
| **Husband's occupation** | Unemployed | 0.239 | 0.0024 | 0.0006 | 0.4168 |
| | Farmer | **Reference** | | | |
| | Hard worker | −0.0151 | 0.0137 | −0.0002 | −0.1512 |
| | Service | 0.0842 | 0.0151 | 0.0013 | 0.9265 |
| | Business | 0.1809 | 0.0154 | 0.0028 | 2.0342 |
| **Women's occupation** | Housewife | 0.0269 | 0.0907 | 0.0024 | 1.7792 |
| | Others | **Reference** | | | |
| **Media access** | No access | **Reference** | | | |
| | Have access | 0.2009 | 0.0617 | 0.0124 | 9.0405 |
| **History of pregnancy losses** | No records | **Reference** | | | |
| | Have records | −0.0061 | 0.0257 | −0.0002 | −0.1138 |

*(Continued)*

Table 3. (Continued)

| Study Variable | Categories | CI | Elasticity | Contribution to CI | |
|---|---|---|---|---|---|
| | | | | AC | PC |
| **ANC by qualified person** | Inadequate | **Reference** | | | |
| | Adequate | 0.2121 | 0.0871 | 0.0185 | 13.469 |
| **Women's age group** | <20 years | −0.0493 | −0.0164 | 0.0008 | 0.5877 |
| | 20-29 years | −0.0046 | −0.0155 | 0.0001 | 0.0523 |
| | 30-39 years | 0.0404 | 0.0143 | 0.0006 | 0.4208 |
| | 40-49 years | **Reference** | | | |
| **Age at first birth** | <18 years | −0.1138 | −0.005 | 0.0005 | 0.3504 |
| | 18 + years | **Reference** | | | |

**Explained CI= 0.1369; Residual CI= 0.0059**

**CI= Concentration Index; AC= Absolute Contribution; PC= Percentage contribution**

for healthcare, and therefore, women with fewer children are more likely to seek SBA. This emphasizes the importance of family planning programs that enable smaller family sizes, improve access to healthcare and reduce maternal health risks. Media access, highly concentrated among wealthier women (PC = 9.04%) also played a significant role in socioeconomic inequality of SBA usage, highlighting the power of media in raising awareness about maternal health and the importance of SBA. In Bangladesh, media campaigns have proven effective in educating the population about health issues, and increasing media exposure, especially in rural areas, can lead to higher SBA utilization.

Finally, adequate ANC visits during pregnancy showed a 13.47% contribution to CI, meaning it is highly concentrated among wealthier women in access to SBA utilization. An adequate ANC visit could ensure women seek better healthcare compared to others. The overall explained concentration index (CI = 0.1369) showed that there exists pro-rich socioeconomic inequality in access to SBA in Bangladesh. The residual concentration index (CI = 0.0059) indicated that while the socioeconomic factors in the study explain a significant portion of the inequality, there remains a small unexplained portion, which could be influenced by other factors not captured in this analysis. The decomposition analysis clearly identified the key socioeconomic determinants of SBA utilization, with wealth, education, media access, family size, and regional disparities playing substantial roles. The findings emphasized the need for targeted interventions to address socioeconomic inequalities and ensure that SBA services are accessible to all women, particularly in rural and low-income populations, to improve maternal health outcomes and reduce maternal and child mortality in Bangladesh.

Fig 3 presents the CC for SBA use across different living regions in Bangladesh. As a visual representation of inequality in access to SBA services, the curve plots the cumulative share of the population arranged from poorest to richest on the x-axis against the cumulative share of SBA help on the y-axis. The curve for each division lies below the line of equality (45-degree diagonal line), indicating that SBA assisted childbirth was more concentrated among women from the richest households. The Mymensingh and Sylhet divisions showed the largest gap between the lines of equality, indicating the highest level of inequality in access to an SBA during childbirth existed in these regions of Bangladesh. This gap gradually narrows from Chattogram to Rangpur and Barishal; the lines of these three regions were very close, which indicated a similar level of inequality existed in these regions. The Khulna division had the smallest disparity, indicating more equitable access to SBA services among socioeconomic classes. Combining CC with map in Fig 2 (SBA coverage by division), we noticed a notable pattern in which the gap between the CC and the line of equality narrows as the overall SBA coverage percentage increases. Divisions with lower SBA coverage, such as Mymensingh and Sylhet, showed bigger inequality gaps, but those with higher coverage, such as Khulna, Dhaka, and Rajshahi, showed narrower gaps, indicating more equitable access to SBA across socioeconomic levels.

 

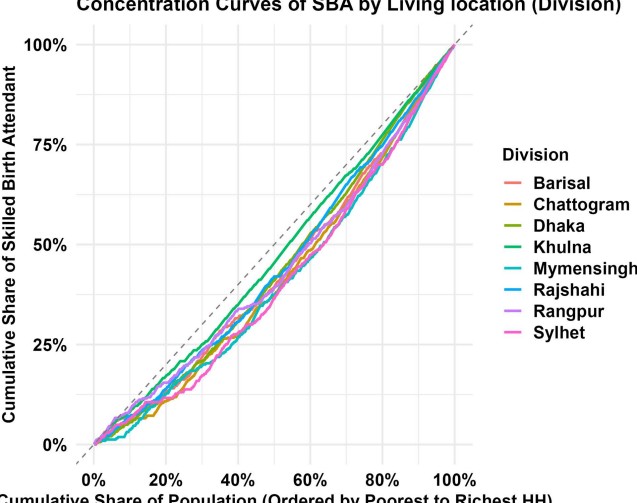

**Fig 3. Concentration curves for estimating divisional disparities in SBA assisted childbirth.**

## Results of machine learning approaches

The varying aptitudes of discrete ML paradigms in predicting SBA-assisted childbirths were assessed across different pivotal metrics, including accuracy, precision, recall, F1 score, AUC and specificity, as displayed in Table 4.

Fig 4 depicts bar charts contrasting the effectiveness measures for multiple ML approaches utilized to predict SBA utilization. The metrics presented a comprehensive evaluation of the models' performance. The bar charts showed that ANN performed exceptionally well with the highest AUC. LR, LGBM, CATB, XGBoost and SVM also displayed robust outcomes with a consistent score of AUC.

The ROC curve (Fig 5) for 10 unique ML models confirmed that ANN and LR models showed the strongest performance, each with an AUC of approximately 0.81 and 0.80. This indicates the outperformance of these models, which is consistent with the findings from Table 4. These models consistently had curves that are closer to the top-left corner,

**Table 4. Results of model performance on test dataset.**

| Model | Accuracy | Recall | Specificity | Precision | F1 score | AUC (95% CI) |
|---|---|---|---|---|---|---|
| **LR** | 0.74 | 0.73 | 0.76 | 0.84 | 0.78 | 0.80 (0.78, 0.84) |
| **RF** | 0.72 | 0.76 | 0.65 | 0.81 | 0.78 | 0.77 (0.73, 0.80) |
| **SVM** | 0.72 | 0.74 | 0.70 | 0.82 | 0.78 | 0.79 (0.76, 0.82) |
| **GBM** | 0.70 | 0.77 | 0.57 | 0.77 | 0.77 | 0.74 (0.71, 0.78) |
| **XGBoost** | 0.72 | 0.74 | 0.68 | 0.81 | 0.78 | 0.79 (0.76, 0.82) |
| **DT** | 0.69 | 0.69 | 0.68 | 0.81 | 0.74 | 0.75 (0.72, 0.78) |
| **ANN** | 0.74 | 0.74 | 0.75 | 0.84 | 0.79 | 0.81 (0.78, 0.83) |
| **NB** | 0.71 | 0.68 | 0.78 | 0.85 | 0.76 | 0.79 (0.76, 0.82) |
| **LGBM** | 0.73 | 0.74 | 0.72 | 0.83 | 0.78 | 0.79 (0.76, 0.82) |
| **CATB** | 0.73 | 0.75 | 0.69 | 0.82 | 0.78 | 0.79 (0.76, 0.82) |

Note: LR = Logistic Regression; RF = Random Forest; SVM = Support Vector Machine; GBM = Gradient Boosting Machine; XGBoost = Extreme Gradient Boosting; DT = Decision Tree; ANN = Artificial Neural Network; NB = Naïve Bayes; LGBM = Light Gradient Boosting Machine; CATB = CatBoost; AUC = Area Under the Receiver Operating Characteristic Curve; CI = Confidence Interval.

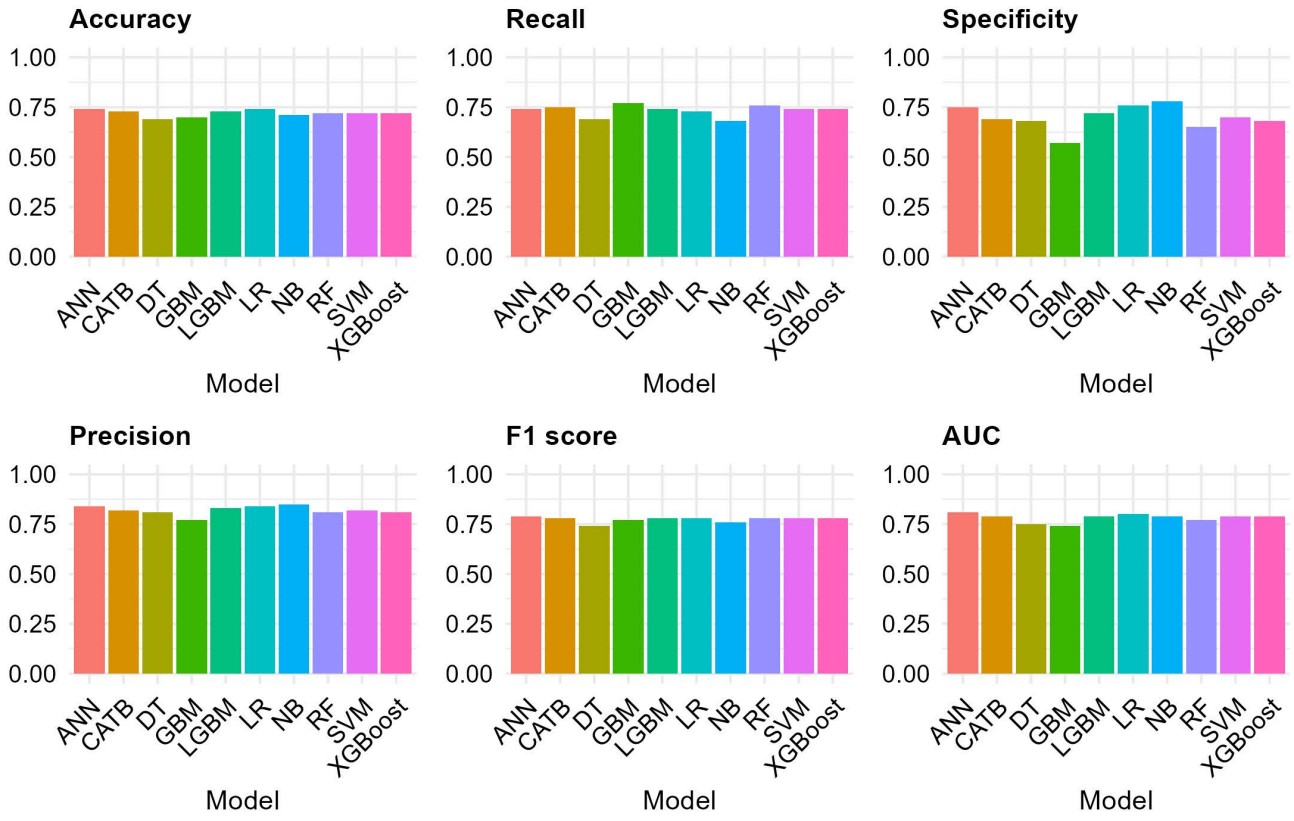

**Fig 4. Model comparison across various metrics.**

indicating a strong ability to discriminate between SBA-assisted and non-assisted births. On the other hand, LGBM, CATB, SVM, XGBoost and NB also performed well with an AUC value of 0.79.

Table 5 represents the statistical pairwise comparison test of the AUC metric between models. The LR model showed significant performance difference between all models except ANN. ANN showed significant difference except for LR and NB. SVM showed no difference from XGBoost, and both of them showed no significant difference with NB. GBM also showed significant performance difference between all models except SVM. LGBM and CATB showed significantly different performance with LR, RF, DT, GBM, and ANN.

In Fig 6, the feature importance of the top four ML models, ANN, LR, LGBM and CATB (LGBM and CATB were chosen based on accuracy because LGBM, CATB, SVM, XGBoost, and NB showed approximately equal AUC values) provides insights into the most influential determinants for an SBA-assisted childbirth, with comparable results across models. In all models, the number of ANC visits before delivery was the predominant feature impacting SBA utilization, underscoring the pivotal role of adequate prenatal care. In every model, additional key aspects included wealth index and living location, demonstrating the importance of socioeconomic status and residential location in determining access to SBA services. Both women's and their husband's education levels also played significant roles to SBA usage. In every model, total ever born children ranked second in feature importance. In MBLR model ANC visit, total ever born children, living location, religion, wealth index and women's education were the top six contributing variables. That's indicated the consistency of results from ML models with the results of the MBLR model. Additionally, media access, religion, women and their husband's occupation and history of pregnancy losses also played a significant role in feature importance, found among the

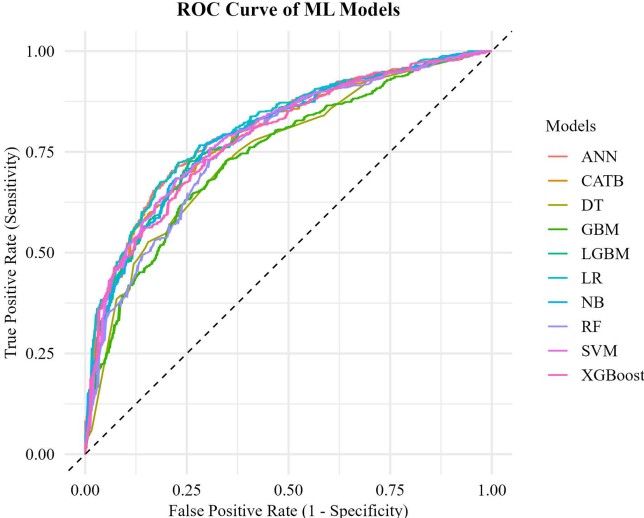

**Fig 5. ROC curve of all ML models.**

**Table 5. Results of DeLong's test for pair wise comparison of AUC in between models.**

| Models | LR | RF | SVM | DT | GBM | XGBoost | NB | ANN | LGBM |
|---|---|---|---|---|---|---|---|---|---|
| **RF** | 0.000*** | | | | | | | | |
| **SVM** | 0.011** | 0.004** | | | | | | | |
| **DT** | 0.000*** | 0.122 | 0.000*** | | | | | | |
| **GBM** | 0.000*** | 0.039** | 0.000*** | 0.724 | | | | | |
| **XGBoost** | 0.004** | 0.050* | 0.276 | 0.002** | 0.000*** | | | | |
| **NB** | 0.046** | 0.031** | 0.985 | 0.002** | 0.001*** | 0.501 | | | |
| **ANN** | 0.152 | 0.000*** | 0.064* | 0.000*** | 0.000*** | 0.014** | 0.176 | | |
| **LGBM** | 0.011** | 0.003** | 0.746 | 0.000*** | 0.000*** | 0.414 | 0.834 | 0.048** | |
| **CATB** | 0.008** | 0.018** | 0.501 | 0.001*** | 0.000*** | 0.680 | 0.665 | 0.034** | 0.713 |

All values in the table represent *p-values* for pairwise comparisons of AUC in between models.

*** for *p-value* ≤ 0.01, ** for *p-value* ≤ 0.05 and * for *p-value* ≤ 0.10

higher position features of several models, particularly in LGBM. In addition, women's age group, relation with house-hold head, age at first birth and household size showed very little importance among these ML models. The MBLR also showed insignificant results (*p*-value > 0.05) for these variables.

Fig 7 presents the SHAP waterfall plots that show the ANN model's prediction of SBA-assisted childbirths, with a focus on the likelihood of utilization dependent on various influencing factors. The prediction illustrated that the features: adequate ANC visit, women who were housewives, rich wealth index, higher education, having media access and having a history of pregnancy losses showed a positive impact on probability prediction for an SBA assisted delivery. On the other hand, the features inadequate ANC visit, poor wealth index, having 2 or 3 + children, no records of pregnancy losses, lower education level of husband, women who are not housewives, no media access, and living in rural area showed a negative impact on the probability for an SBA assisted childbirth. The results from the waterfall plot in consistent with the AORs of the MBLR model. In the figure, the probability of an SBA assisted delivery increased from the base probability of

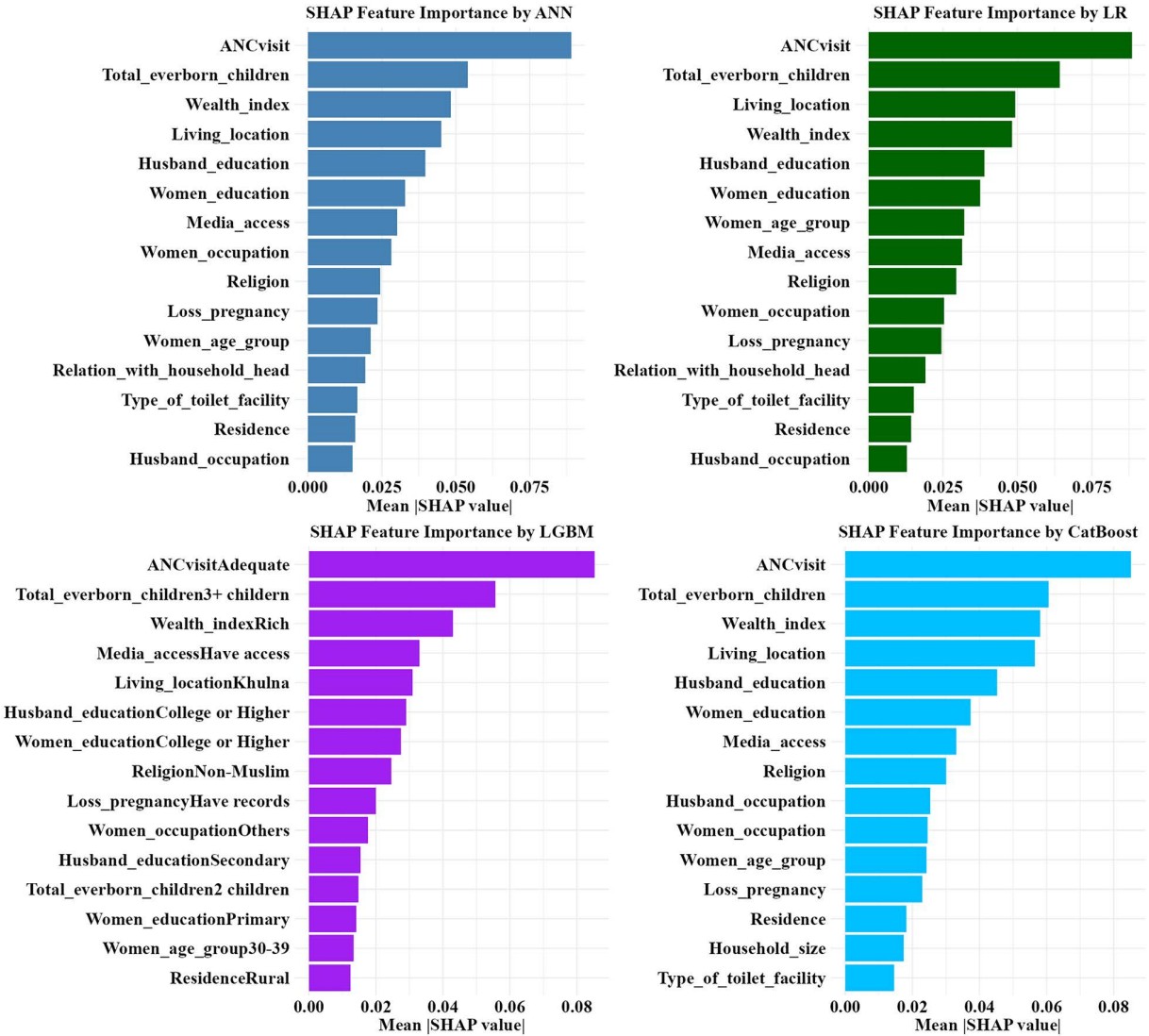

**Fig 6. SHAP bar plots showing most influential determinants of SBA usage.**

0.503 to 0.764 for the woman whose row id is 51 (Fig 7A). Similarly for row id of 124, the probability decreased to 0.423 (Fig 7B).

In Fig 8, the SHAP Global Feature Importance (beeswarm) plot (Fig 8A) represents the directional impact of the determinants in model interpretation, which highlighted that a higher number of ANC visits, a higher wealth quintile, a higher level of education for the respondent and her partner, and having media access had a positive effect on SBA utilization during childbirth. The Feature Interaction (SHAP dependency plot) (Fig 8B) employed to discover the effect of the two most important determinants of SBA. The plot demonstrated that adequate ANC was concentrated among wealthier women and a different scenario among the poorer women. This disparity indicates a large socioeconomic gap in ANC of women, which results in a higher concentration of SBA among women from richer households.

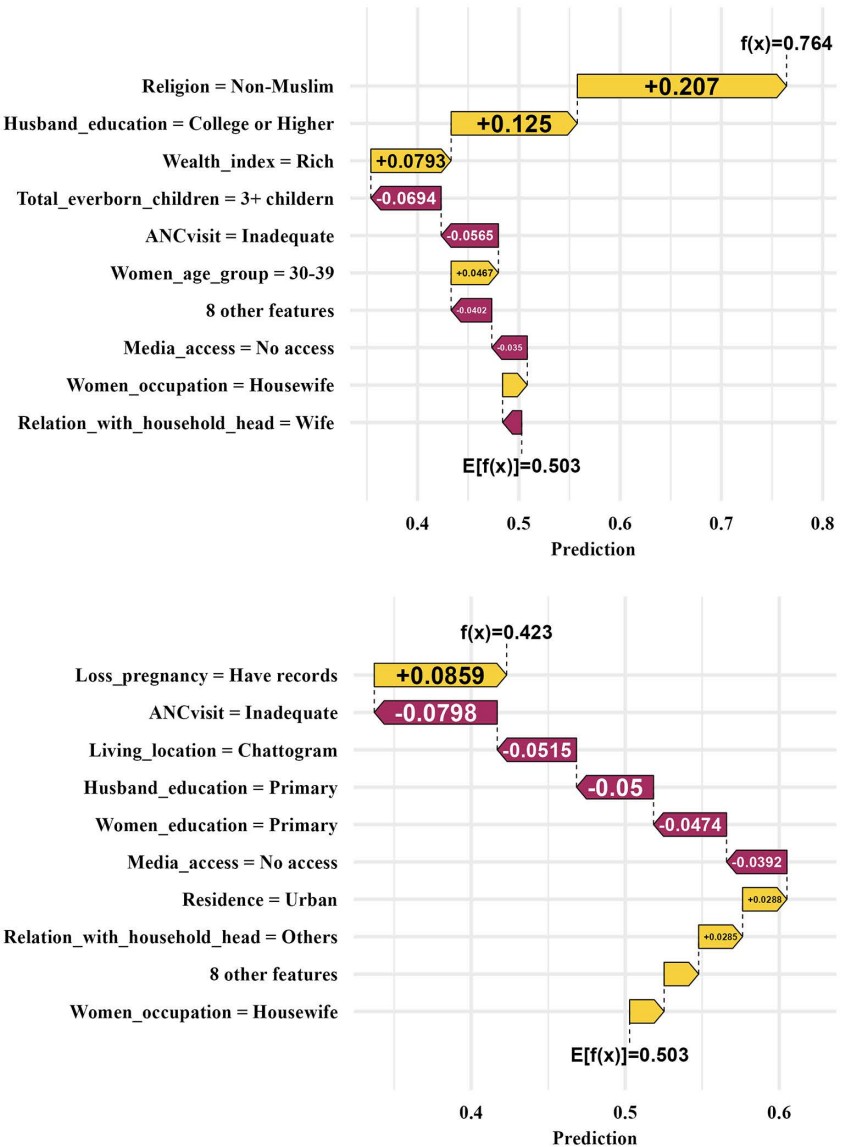

**Fig 7. SHAP waterfall plot for demonstrating individual instance.** A. For row id 51. B. For row id 124.

## Discussion

This study investigated the determinants and inequalities associated with SBA utilization during childbirth among Bangladeshi women. By applying MBLR and a range of ML models, the research aimed to enhance predictive understanding of SBA use and identify key socio-demographic and economic factors influencing it. The study also employed spatial mapping, Wagstaff decomposition, and concentration curves to visualize and quantify regional and socioeconomic disparities in SBA utilization. Although ML applications in maternal health had been explored in African contexts [13,33], such approaches were relatively unexplored in Bangladesh, with only a few studies addressing inequality and one using WHO's HEAT software with earlier BDHS data [24,27]. The national SBA usage rate was found to be 65.43%, showing an improvement over previous estimates [21,22,24,28], yet still lagging behind regional neighbors such as Maldives (100%),

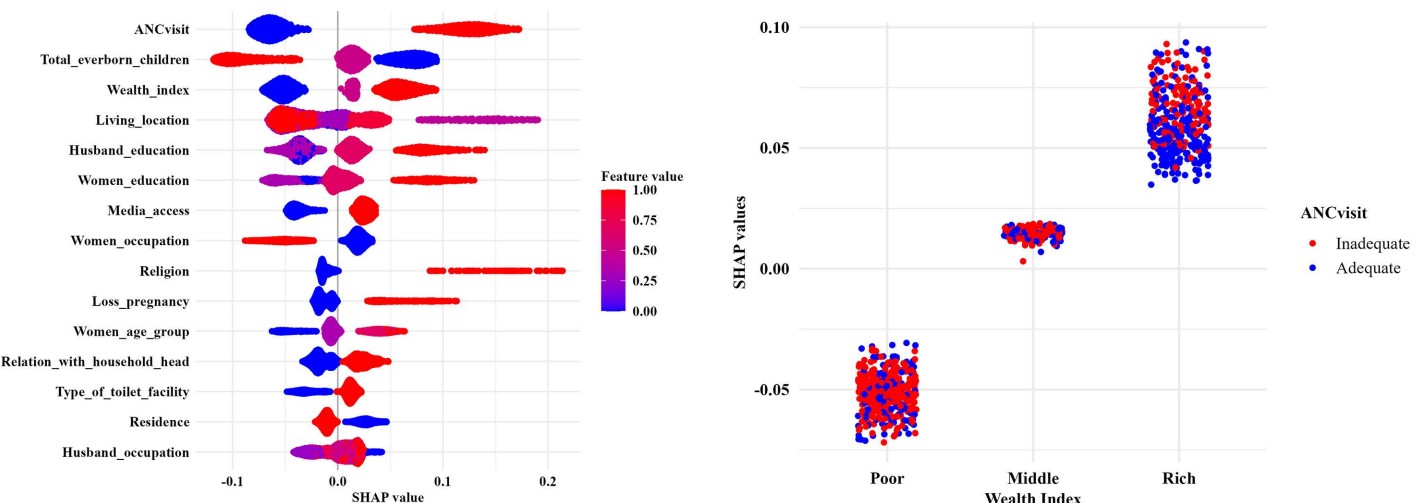

**Fig 8. SHAP global feature importance and interaction of key factors affecting SBA usage.** A. SHAP beeswarm plot. B. SHAP dependency plot.

India (89%), Nepal (80%), and Pakistan (69%) [57–60]. Significant geographic differences persist within Bangladesh, with Khulna Division reporting the highest SBA usage at 84%, followed by Dhaka (72%). In contrast, Mymensingh (54%) and Sylhet (54.8%) showed significantly lower rates. These disparities confirm findings from previous BDHS-based studies [24,27] and are also evident in countries like India and several Sub-Saharan African nations [61,62]. The continued inadequacy of maternal health infrastructure, especially in rural and hard-to-reach areas, remains a critical challenge. Interventions such as the "Skilled Health Entrepreneur" initiative in Sylhet, which trains and deploys community-based birth attendants, offer scalable models for addressing these regional disparities [25,26].

Among the ML models tested, the ANN demonstrated the highest predictive accuracy (74%), recall (0.74), specificity (0.75), precision (0.84), f1-score (0.79) and AUC (0.81). The metrics suggested a balanced and robust predictive performance of the model. The recall value implied that 74% of women were clearly identified as requiring SBA, but the other 26% false negative represented missed opportunities of SBA, which might help to increase the risk of preventable neonatal and maternal complications during childbirth. In contrast, 84% of women who needed SBA were correctly classified, supporting a good use of limited healthcare resources. The specificity value also indicated a valuable ability to exclude low risk cases. The successful application of the model can make things better by making predictions more accurately, capturing complex interactions, and allowing for risk classification that can be acted on, though the use of classical regression model was still useful for figuring out what causes things and making sense of them.

Chi-square testing initially showed that the sex of the household head was statistically insignificant—a result that contrasts with studies in Sub-Saharan Africa [63] and Madagascar [64], where the gender of the household head significantly influenced SBA usage. After excluding this variable, MBLR and SHAP analysis across models like ANN, LR, LGBM and CATB consistently identified top determinants of SBA usage: the number of ANC visits, household wealth, living location, both maternal and paternal education levels, total children ever born, media access, religion, and history of pregnancy loss. These results aligned with previous Bangladesh-based studies [21] and similar international research [13,33]. Women residing in Khulna, Rajshahi, and Dhaka divisions were found to have significantly higher odds of SBA utilization compared to those in Mymensingh, reflecting long-standing geographical disparities reported in BDHS 2017–18 analyses [21,28,34]. Urban residency was also associated with greater SBA use, a finding consistent with global maternal health literature [14,15,19,64]. These urban-rural disparities are rooted in differences in service availability, infrastructure, and socio-economic opportunities [12,23,60,65,66].

Education, both maternal and spousal, emerged as a strong predictor of SBA use, with women and their husbands who had college or higher education levels more likely to opt for SBA. This finding is consistent with previous studies in Bangladesh and similar settings [14,21,23,34,63]. Higher education enhances awareness, confidence in health systems, and decision-making autonomy. A 2024 systematic review also affirmed that men with secondary or higher education levels were supportive of skilled delivery services [67]. Additionally, women from wealthier households exhibited higher SBA usage, reflecting better affordability and access to maternal services, consistent with studies in Bangladesh and LMICs such as Nigeria, Ethiopia, and Indonesia [19,64]. Homemakers also showed higher SBA use than working women, likely due to increased availability for ANC visits and greater autonomy over childbirth planning [23,66,68]. Furthermore, women whose husbands were engaged in business were more likely to use SBA services, likely due to higher socio-economic status and better health awareness [21,22,34]. Women with access to improved sanitation had higher SBA usage, as poor hygiene conditions are linked to increased maternal mortality [69]. Interestingly, Muslim women were found to have lower SBA utilization compared to their non-Muslim counterparts, a trend observed in both Bangladesh and parts of Africa [21,22,63], though opposite patterns were seen in Madagascar [64]. Cultural norms, conservative practices, and restricted female mobility may contribute to these disparities [22].

ANC visits were the most consistent determinant of SBA use across all ML models, with women receiving recommended ANC significantly more likely to seek skilled delivery services [19,21,23,34]. ANC visits enhance knowledge, risk detection, and trust in the health system [68,70]. Access to mass media was also an influential factor, promoting health-seeking behaviors and increasing awareness of maternal services [28,71]. Women with fewer children were more likely to use SBA, possibly due to increased caution, younger age, or prior complications [21,63]. Additionally, women with a history of pregnancy loss were more likely to seek SBA services, reflecting heightened risk perception [72]. The study also found significant socio-economic inequalities in SBA utilization, with a higher concentration of SBA use among wealthier women in nearly all regions of Bangladesh. The Wagstaff decomposition revealed that wealth index, ANC visits, education (both maternal and paternal), and media exposure were major contributors to the inequality. These findings align with inequality studies from Bangladesh and Ghana [20,24,43,73]. Regional inequalities were most pronounced in the Khulna, Dhaka, and Rajshahi division, as well as in urban areas, highlighting both spatial and social gaps in SBAs coverage. To address these disparities, targeted interventions are needed in rural and low-income regions, including improved health infrastructure, deployment of trained personnel, and financial incentives. Promoting female education, expanding ANC coverage, and utilizing media for awareness campaigns can empower women to seek skilled care. Culturally sensitive community-based strategies and the engagement of local leaders will also be crucial to overcoming geographic and socio-cultural barriers to SBA utilization.

## Strengths and limitations

This study has several notable strengths. It utilizes the most recent nationally representative BDHS 2022 dataset, ensuring strong external validity and relevance for contemporary maternal health policy. The integration of traditional statistical modeling (MBLR), socioeconomic inequality analysis (Wagstaff decomposition and CC), and advanced ML techniques provides a comprehensive and methodologically robust framework for understanding SBA utilization. The use of interpretable ML tools, particularly SHAP analysis, enhances transparency by identifying the relative importance and directional effects of key determinants, bridging the gap between predictive accuracy and policy relevance. Additionally, spatial and regional analyses allow the identification of geographic disparities and high-risk areas, providing actionable insights for targeted interventions.

Despite these strengths, several limitations should be acknowledged. First, the cross-sectional design of the BDHS data limits causal interpretation between the identified predictors and SBA utilization. Second, reliance on self-reported information may introduce recall or reporting bias, particularly for antenatal care visits and delivery assistance. Third, important health system–level factors—such as quality and continuity of care, distance to health facilities, and provider availability—were not captured in the dataset and could not be examined.

Future studies using longitudinal data and incorporating service quality and health system indicators would help address these limitations and further strengthen evidence on equitable access to skilled delivery care in Bangladesh.

**Policy recommendations**

The findings of this study can generate several evidence-based policy recommendations. First, expanding access to high-quality and adequately staffed ANC services should be a national priority, as ANC visits emerged as the most consistent and influential determinant of SBA utilization across multilevel regression, machine learning models, and SHAP-based interpretability analyses. Strengthening ANC coverage—particularly through qualified providers—can serve as a gateway to increasing SBA-assisted deliveries.

Second, region-specific and geographically targeted interventions are urgently needed in low-performing divisions such as Mymensingh and Sylhet, where both SBA coverage and equity remain poor. These interventions should focus on improving rural health infrastructure, deploying trained and motivated skilled personnel, and enhancing transportation and referral systems to reduce geographic barriers to facility-based delivery of care.

Third, given the strong pro-rich inequality in SBA utilization identified through Wagstaff decomposition, pro-poor financing mechanisms—such as conditional cash transfers, maternal health vouchers, and free or subsidized delivery services—should be expanded to reduce financial barriers faced by disadvantaged women, particularly in rural and low-income households.

Fourth, sustained investments in female education and mass media–based health communication strategies are essential, as both education and media exposure significantly increase the likelihood of SBA usage. Targeted awareness campaigns can improve health literacy, promote timely care-seeking behavior, and reinforce the benefits of skilled delivery services.

Finally, culturally sensitive, community-based approaches are needed to address sociocultural barriers identified in the study, including religious and household-level influences on SBA utilization. Engaging local leaders, religious figures, and male partners can enhance community acceptance and support for skilled maternal healthcare services.

Together, these policy actions directly address the key socioeconomic, geographic, and behavioral determinants identified in this study and are critical for reducing maternal and neonatal mortality and achieving equitable, sustainable maternal healthcare coverage in Bangladesh.

## Conclusion

This study comprehensively examined the factors influencing SBA utilization during childbirth in Bangladesh, revealing significant socioeconomic and geographical disparities. Although the overall SBA coverage reached 65.43%, its distribution remains uneven across regions. Women in urban areas, those from wealthier households, and those with higher education were significantly more likely to receive skilled care at delivery, highlighting persistent inequities in maternal healthcare access. These disparities underscore the need for evidence-based, regionally targeted interventions to ensure equitable coverage.

Using an integrated analytical framework that combined descriptive analysis, MBLR, and ML techniques, the study identified key determinants of SBA utilization. ANC visits, household wealth, place of residence, education of women and their spouses, parity, media exposure, religious background, and prior pregnancy loss consistently emerged as influential factors. ML models, particularly ANN, LR, LGBM, and CATB, demonstrated strong predictive performance. SHAP-based interpretability provided nuanced insights into how these factors individually and collectively shape maternal healthcare-seeking behavior, bridging the gap between predictive modeling and policy-relevant understanding.

Future research should adopt longitudinal designs to better capture causal relationships and to explore additional dimensions, such as health system capacity, cultural norms, and the quality and continuity of maternal care. Improving SBA coverage in Bangladesh requires addressing entrenched socioeconomic and geographic barriers by expanding ANC

services, improving health infrastructure, empowering education, and targeting media outreach. Ensuring universal access to SBAs is essential to advancing maternal and neonatal health, promoting equity, and achieving the SDGs.

## Supporting information

**S1 Table. Pseudocode for SHAP implementation and decomposition setup.**
(DOCX)

**S2 Table. Details information about model hyperparameters and required packages.**
(DOCX)

## Acknowledgments

The authors gratefully acknowledge the Bangladesh Demographic and Health Survey (BDHS-2022) and the National Institute for Population Research and Training (NIPORT) of the Ministry of Health and Family Welfare, Dhaka, Bangladesh, for providing access to the dataset used in this study. ChatGPT 4.0 was used for better readability and language of the manuscript.

## Author contributions

**Conceptualization:** Rafayet Rahman Ridoy, Rehana Parvin.

**Data curation:** Rafayet Rahman Ridoy, Rehana Parvin, Tofayel Ahmed, Tahira Mahbub.

**Formal analysis:** Rafayet Rahman Ridoy, Rehana Parvin, Tofayel Ahmed, Tahira Mahbub.

**Investigation:** Rafayet Rahman Ridoy, Rehana Parvin, Tahira Mahbub.

**Methodology:** Rafayet Rahman Ridoy, Rehana Parvin, Tofayel Ahmed, Tahira Mahbub.

**Project administration:** Rafayet Rahman Ridoy, Rehana Parvin.

**Resources:** Rafayet Rahman Ridoy, Rehana Parvin, Tahira Mahbub.

**Software:** Rafayet Rahman Ridoy, Rehana Parvin, Tofayel Ahmed, Tahira Mahbub.

**Supervision:** Rafayet Rahman Ridoy, Rehana Parvin, Tofayel Ahmed.

**Validation:** Rafayet Rahman Ridoy, Rehana Parvin, Tofayel Ahmed.

**Visualization:** Rafayet Rahman Ridoy, Rehana Parvin, Tahira Mahbub.

**Writing – original draft:** Rafayet Rahman Ridoy, Rehana Parvin, Tofayel Ahmed, Tahira Mahbub.

**Writing – review & editing:** Rafayet Rahman Ridoy, Rehana Parvin, Tofayel Ahmed.

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
