## [Decision Letter · Decision Letter 0]

10 Dec 2025

PONE-D-25-44386Determinants, demographic and socio-economic inequalities in skilled birth attendants during childbirth in Bangladesh: a combined study of machine learning and decomposition analysisPLOS One

Dear Dr. Parvin,

Thank you for submitting your manuscript to PLOS ONE. After careful consideration, we feel that it has merit but does not fully meet PLOS ONE’s publication criteria as it currently stands. Therefore, we invite you to submit a revised version of the manuscript that addresses the points raised during the review process.

We look forward to receiving your revised manuscript.

Kind regards,

Md. Obaidur Rahman, Ph.D.

Academic Editor

PLOS One

Additional Editor Comments (if provided):

Reviewers' comments:

Reviewer's Responses to Questions

**Comments to the Author**

1. Is the manuscript technically sound, and do the data support the conclusions?

Reviewer #1: Partly

Reviewer #2: Partly

2. Has the statistical analysis been performed appropriately and rigorously? 

Reviewer #1: Yes

Reviewer #2: Yes

3. Have the authors made all data underlying the findings in their manuscript fully available?

Reviewer #1: Yes

Reviewer #2: Yes

4. Is the manuscript presented in an intelligible fashion and written in standard English?

Reviewer #1: Yes

Reviewer #2: Yes

5. Review Comments to the Author

Reviewer #1: This manuscript, entitled “Determinants, demographic and socio-economic inequalities in skilled birth attendants during childbirth in Bangladesh: a combined study of machine learning and decomposition analysis” addresses a public health issue with a study design that integrates multilevel regression, machine learning, and inequality decomposition, offering interesting policy insights. It leverages nationally representative survey data from Bangladesh, applies classical ML techniques, and employs SHAP for model interpretation. The study provides valuable findings and generates actionable evidence for the government and policymakers. However, the manuscript has notable weaknesses, including unclear writing, insufficient detail on ML tuning and validation, usage of outdated ML models, limited novelty (as ML largely confirms regression results), underinterpretation of findings, and gaps in reproducibility.

Strengths

1) The study tackles a critical issue of maternal and neonatal health by focusing on skilled

birth attendant (SBA) utilization in Bangladesh.

2) The integration of multilevel logistic regression, multiple machine learning models, and Wagstaff decomposition is methodologically ambitious.

3) Inclusion of SHAP-based feature importance and waterfall plots improves interpretability of ML results.

4) Wagstaff decomposition and concentration index analysis are carefully executed, with wealth and ANC visits identified as top contributors to inequality

Weaknesses:

1) Several sentences are grammatically incorrect or poorly phrased (e.g., “did not able to achieve”), which detracts from professionalism. Terminology inconsistencies (metrics vs “matrices”) also reduce clarity.

2) The study relies primarily on older, classical ML techniques (e.g., logistic regression, SVM, naïve Bayes, decision tree, random forest, gradient boosting). More advanced state-of-the-art methods, such as LightGBM, CatBoost, TabNet, and deep learning architectures like transformer-based models (TabTransformer, FT-Transformer), could potentially capture richer feature interactions and outperform the models used here.

3) The reported model accuracies (~72%) and AUC (~0.80) are relatively low for a binary classification task. While statistically informative, such performance limits the utility of the models for reliable real-world prediction and intervention targeting.

4) The authors addressed missing values by removing all cases with incomplete data. This approach is not considered best practice in data science, as it can reduce the effective sample size, introduce selection bias, and compromise the generalizability of the results.

5) No clear description of hyperparameter tuning for complex models. Without this, performance results may not reflect best-case capabilities.

6) Reliance on a single 70/30 train-test split without cross-validation weakens claims about robustness. Performance could vary with different splits, and no confidence intervals or significance tests are provided.

7) Performance trade-offs between models (e.g., high recall but low specificity in Random Forest) are not analyzed. No discussion of why ANN underperformed or potential non-linear interactions that ML might uncover beyond regression.

8) While data are available, no code or pseudocode is shared. Complex steps (e.g., SHAP implementation, decomposition setup with survey weights) may be difficult to replicate without further documentation.

Recommendations

1) Authors should provide details of hyperparameters, ANN structure, and training methods. If default settings were used, state this explicitly. Ideally, rerun models with tuned parameters and update results.

2) Authors should use k-fold cross-validation or bootstrap methods to report stable averages and confidence intervals for metrics like AUC. Report statistical tests (e.g., DeLong’s test) to check significance of performance differences between models.

3) Authors should explore SHAP dependence or partial dependence plots to show non-linear relationships. Discuss trade-offs between precision and recall in practical terms (e.g., implications of false negatives in maternal health contexts). Highlight unique contributions of ML beyond regression.

4) Authors should provide supplementary code snippets, pseudocode, or detailed appendices describing analytical steps (e.g., specific R/Stata commands). This will allow others to build upon the work and increase transparency.

5) For data samples having missing values, authors should use some imputation techniques, such as SMOTE, or ADASYN.

6) The authors should consider experimenting with state-of-the-art models, which may provide stronger predictive performance and deeper insights than the classical methods currently used.

7) The manuscript would benefit from thorough proofreading to improve clarity, grammar, and overall readability.

Reviewer #2: Determinants, demographic and socio-economic inequalities in skilled birth attendants during childbirth in Bangladesh: a combined study of machine learning and decomposition analysis

Manuscript Number: PONE-D-25-44386

1. The Title of the Manuscript is very long. I suggest making it smaller and compact.

2. At line 199, what is MSBA, its not used elsewhere in paper.

3. The authors did not report the confusion matrix.

4. Since the accuracy is almost nearby for all models. Whether Hyperparameter tuning, cross validation is performed or not?

5. The authors did not report about parameters choice, overfitting or underfitting of model.

6. What were the hyperparameters and how were they tuned?

7. It is unclear whether performance differences between models are statistically significant.

8. It’s not mentioned that after applying SMOTE how many cases are considered for further evaluation. Also, When SMOTE has applied after or before split?

9. A Beeswarm SHAP plot can be added.

10. Its better if authors can provide a research flow and Overview of the machine learning framework applied for better understanding to readers.

11. The conclusion is little longer. It’s better to keep strengths and limitations as separate section.

12. Based on this study what are the possible recommendations.

13. The Abstract is too long, weak Conclusion and Introduction section lacking literature review on ML and XAI applications for SBA prediction. The methodology section did not mentioned utilization of XAI.

14. For the present study, How and why the initial independent variables are selected.

15. A minor English editing is required.

6. PLOS authors have the option to publish the peer review history of their article (what does this mean?). If published, this will include your full peer review and any attached files.

Reviewer #1: No

Reviewer #2: No

---

## [Author Response · Author response to Decision Letter 1]

12 Jan 2026

Manuscript Number: PONE-D-25-44386

Replies to the Reviewers’ Comments

The authors would like to thank the respected editorial manager and reviewers for providing valuable comments, constructive criticisms and useful suggestions that have greatly improved the quality, presentation and style of the manuscript. The authors have done their level best to address all the comments raised by the respected reviewers in the revised manuscript. The text edited in the revised manuscript is typed in red colour.

The point-by-point replies to the respected editorial manager and reviewers’ comments are as follows:

Responses to Editorial Manager:

Reply: Thank you for your valuable suggestion. We formatted the revised manuscript without track change file named “Manuscript” following the submission guidelines of PLOS ONE.

2. Please note that PLOS One has specific guidelines on code sharing for submissions in which author-generated code underpins the findings in the manuscript. In these cases, we expect all author-generated code to be made available without restrictions upon publication of the work. Please review our guidelines at https://journals.plos.org/plosone/s/materials-and-software-sharing#loc-sharing-code and ensure that your code is shared in a way that follows best practice and facilitates

reproducibility and reuse.

Reply: Thank you for your suggestion. We will be happy and have no issues on sharing the code and the data used in our study.

Reply: Thank you so much for the support. The joint decision made by the authors is that they have no issues for sharing the data publicly.

Reply: Thank you. We have updated the manuscript adding the ethical statement on the methodology part of the paper.

Reply: Thank you for your valuable suggestion. We have added a separate caption for each figure in the manuscript.

Reply: Thank you for your comment. We read the reviewers recommendations very carefully, and they did not mention any specific works for citation.

Responses to Reviewer 1:

Recommendations

1) Authors should provide details of hyperparameters, ANN structure, and training methods. If default settings were used, state this explicitly. Ideally, rerun models with tuned parameters and update results.

Reply: Thank you. Based on your valuable suggestion we have modified the model’s results. In past default model structure was used. Now we updated the results by tuning best hyperparameter using 5-fold cross validation (Table 4). And we used random search for finding the best hyperparameters with a tune length of 10. Also details about ANN structure has added in the Methodology section.

2) Authors should use k-fold cross-validation or bootstrap methods to report stable averages and confidence intervals for metrics like AUC. Report statistical tests (e.g., DeLong’s test) to check significance of performance differences between models.

Reply: Thank you for your suggestion. We have updated the result according to your valuable suggestion. We have used 5-fold CV and reported AUC with 95%CI (Table 4). Also, we have performed DeLong’s Test (Table 5) for checking significance of performance differences between models.

3) Authors should explore SHAP dependence or partial dependence plots to show non-linear relationships. Discuss trade-offs between precision and recall in practical terms (e.g., implications of false negatives in maternal health contexts). Highlight unique contributions of ML beyond regression.

Reply: As per your valuable recommendation a SHAP Dependence plot has added (figure 8). The discussion about the model matrices of ANN model have been discussed in maternal health contexts in Discussion part.

4) Authors should provide supplementary code snippets, pseudocode, or detailed appendices describing analytical steps (e.g., specific R/Stata commands). This will allow others to build upon the work and increase transparency.

Reply: Thank you. According to your valuable suggestion some commands and name of library functions have added in methodology section.

5) For data samples having missing values, authors should use some imputation techniques, such as SMOTE, or ADASYN.

Reply: Thank you for your comment. We also believe that exclusion is not an ideal way. We have a large data set and the variable “number of ANC visit” had 250 women’s information missing (also have missing information on other variables). That’s why we preferred to exclude these sample.

6) The authors should consider experimenting with state-of-the-art models, which may provide stronger predictive performance and deeper insights than the classical methods currently used.

Reply: We thank the reviewer for the suggestion. Accordingly, we implemented a state-of-the-art model LightGBM and CatBoost using 5-fold CV with hyperparameters. The LightGBM model achieved accuracy of 0.73, recall of 0.74, specificity of 0.72, precision of 0.83, f1-score of 0.78 and AUC of 0.79, on the other hand CatBoost achieved accuracy of 0.73, recall of 0.75, specificity of 0.69, precision of 0.82, f1-score of 0.78 and AUC of 0.79, which are comparable to the best models in our study.

Due to limitations in the library support within R 4.5.1 the additional advanced models could not be reliably implemented. We have incorporated these results of LightGBM and CatBoost into the revised manuscript and clarified this point in Methodology and Result Section.

7) The manuscript would benefit from thorough proofreading to improve clarity, grammar, and overall readability.

Reply: Thank you so much for your comment. Considering your valuable recommendation, we revised the paper and did correction to improve readability.

Responses to Reviewer 2:

1. The Title of the Manuscript is very long. I suggest making it smaller and compact.

Reply: Thank you. As per your valuable recommendation, we proposed a new title for the manuscript, as follows:

“Determinants and disparities in skilled birth attendants during childbirth in Bangladesh: a study of machine learning and decomposition analysis”.

2. At line 199, what is MSBA, it’s not used elsewhere in paper.

Reply: Thank you. It was a mistake. Please accept our apologies. The texts are being correctly written.

3. The authors did not report the confusion matrix.

Reply: As per your suggestion, we have reported it on methodology part along with how they created. Also, the model metrics which were calculated from model’s confusion matrix of each model. We used test data set for predicting and creation of confusion matrix.

4. Since the accuracy is almost nearby for all models. Whether Hyperparameter tuning, cross validation is performed or not?

Reply: Thank you for your suggestion. The authors had taken this recommendation seriously. In past default model structure was used. Now we updated the results by tuning best hyperparameter using 5-fold cross validation. And we used random search for finding best hyperparameters. The results have shown in table 4.

5. The authors did not report about parameters choice, overfitting or underfitting of model.

Reply: Thank you. In past default model structure was used. Now we updated the results by tuning best hyperparameter using 5-fold cross validation. And we used random search for finding the best hyperparameters. We justified the performance of model based on AUC value for avoiding overfitting and underfitting of model. The models performances reported in Table 4.

6. What were the hyperparameters and how were they tuned?

Reply: Thank you for your comment. In past default model structure was used. Now we updated the results by tuning best hyperparameter using 5-fold cross validation. And we used random search for hyperparameters with a tune length of 10.

7. It is unclear whether performance differences between models are statistically significant.

Reply: As per the recommendation of another respected reviewer, we performed DeLong’s test to find out performance difference in between models. The results have shown in Table 5.

8. It’s not mentioned that after applying SMOTE how many cases are considered for further evaluation. Also, When SMOTE has applied after or before split?

Reply: Thank you for your comment. We had applied SMOTE to balance the training dataset only and it was applied after splitting the dataset (Details discussed on Methodology section).

9. A Beeswarm SHAP plot can be added

Reply: Thank you. A Beeswarm SHAP plot has added according to your suggestion. (Figure 8)

10. Its better if authors can provide a research flow and Overview of the machine learning framework applied for better understanding to readers.

Reply: Thank you. Considering your valuable suggestion, a research flow and ML framework is represented in figure 1.

11. The conclusion is little longer. It’s better to keep strengths and limitations as separate section.

Reply: As per your valuable recommendation, the authors separated the strength and limitation.

12. Based on this study what are the possible recommendations.

Reply: Thank you again for your valuable suggestion. Possible recommendations are added after strength and limitation as a separate section.

13. The Abstract is too long, weak Conclusion and Introduction section lacking literature review on ML and XAI applications for SBA prediction. The methodology section did not mentioned utilization of XAI.

Reply: Considering your important recommendation, the authors tried hard to find more literatures of ML application in SBA context. But there were not so many studies found who used ML and SHAP based interpretability in this particular topic. Two published literatures found on African population. (Ref 13 & 32) The Abstract and Conclusion is rewritten based on your suggestion. Also, Details about application of XAI added in the Methodology part.

14. For the present study, How and why the initial independent variables are selected.

Reply: Thank you for your comment. We reviewed published literatures on Skilled Birth Attendant. Based on the review we selected the independent variables which mentioned in methodology section.

15. A minor English editing is required.

Reply: Thank you for your suggestion. We have read the manuscript thoroughly for searching grammatical mistakes to correct those mistakes. Also, we did try to improve the readability by rewriting some texts.

---

## [Editor Report · Decision Letter 1]

22 Jan 2026

PONE-D-25-44386R1Determinants and disparities in skilled birth attendants during childbirth in Bangladesh: a study of machine learning and decomposition analysisPLOS One

Dear Dr. Parvin,

Thank you for submitting your manuscript to PLOS ONE. After careful consideration, we feel that it has merit but does not fully meet PLOS ONE’s publication criteria as it currently stands. Therefore, we invite you to resubmit a revised version of the manuscript that addresses the following 'Additional Editor Comments' raised during the review process.

We look forward to receiving your revised manuscript.

Kind regards,

Md. Obaidur Rahman, Ph.D.

Academic Editor

PLOS One

Journal Requirements:

**Additional Editor Comments:**

The authors did not adequately respond to the weaknesses highlighted by Reviewer 1. Furthermore, although Reviewer 1 explicitly recommended sharing code or detailed analytical scripts (e.g., R or Stata) to ensure reproducibility of the complex analytical procedures, these materials were not provided as supplementary files.

The authors are therefore invited to resubmit a revised manuscript that fully addresses all reviewer comments, together with a detailed, point-by-point response indicating the specific sections and line numbers where revisions have been made, and all required supplementary materials. Failure to adequately address these issues may preclude further consideration of the manuscript.

---

## [Author Response · Author response to Decision Letter 2]

27 Jan 2026

Manuscript Number: PONE-D-25-44386

Replies to the Reviewers’ Comments

The authors would like to thank the respected editorial manager and reviewers for providing valuable comments, constructive criticisms and useful suggestions that have greatly improved the quality, presentation and style of the manuscript. The authors have done their level best to address all the comments raised by the respected reviewers in the revised manuscript. The text edited in the revised manuscript is typed in red colour and line number are mentioned based on “Revised manuscript with track change” file.

The point-by-point replies to the respected editorial manager and reviewers’ comments are as follows:

Responses to Editorial Manager:

Reply: Thank you for your valuable suggestion. We formatted the revised manuscript without track change file named “Manuscript” following the submission guidelines of PLOS ONE.

2. Please note that PLOS One has specific guidelines on code sharing for submissions in which author-generated code underpins the findings in the manuscript. In these cases, we expect all author-generated code to be made available without restrictions upon publication of the work. Please review our guidelines at https://journals.plos.org/plosone/s/materials-and-software-sharing#loc-sharing-code and ensure that your code is shared in a way that follows best practice and facilitates

reproducibility and reuse.

Reply: Thank you for your suggestion. We will be happy and have no issues on sharing the code and the data used in our study.

Reply: Thank you so much for the support. The joint decision made by the authors is that they have no issues for sharing the data publicly.

Reply: Thank you. We have updated the manuscript adding the ethical statement on the methodology part of the paper (line no: 310-315).

Reply: Thank you for your valuable suggestion. We have added a separate caption for each figure in the manuscript. (Methodology section: line 236 & Results section: line 326, 544, 581, 587, 602, 622 & 635)

Reply: Thank you for your comment. We read the reviewers recommendations very carefully, and they did not mention any specific works for citation.

Responses to Reviewer 1:

Recommendations

1) Authors should provide details of hyperparameters, ANN structure, and training methods. If default settings were used, state this explicitly. Ideally, rerun models with tuned parameters and update results.

Reply: Thank you. Based on your valuable suggestion we have modified the model’s results. In past default model structure was used. Now we updated the results by tuning best hyperparameter using 5-fold cross validation (Table 4) (line no: 565 in Results Section). And we used random search for finding the best hyperparameters with a tune length of 10 (line no: 258-261 in Methodology Section).

Also details about ANN structure has added in the Methodology section (line no: 277-279)

2) Authors should use k-fold cross-validation or bootstrap methods to report stable averages and confidence intervals for metrics like AUC. Report statistical tests (e.g., DeLong’s test) to check significance of performance differences between models.

Reply: Thank you for your suggestion. We have updated the result according to your valuable suggestion. We have used 5-fold CV and reported AUC with 95%CI (Table 4) (line no: 565 in Results Section).

Also, we have performed DeLong’s Test (Table 5) for checking significance of performance differences between models (line no: 595 in Results Section), (line no: 298-299 in Methodology Section).

3) Authors should explore SHAP dependence or partial dependence plots to show non-linear relationships. Discuss trade-offs between precision and recall in practical terms (e.g., implications of false negatives in maternal health contexts). Highlight unique contributions of ML beyond regression.

Reply: As per your valuable recommendation a SHAP Dependence plot has added (figure 8). The discussion about the model matrices of ANN model have been discussed in maternal health contexts in Discussion part (line no: 664-674).

4) Authors should provide supplementary code snippets, pseudocode, or detailed appendices describing analytical steps (e.g., specific R/Stata commands). This will allow others to build upon the work and increase transparency.

Reply: Thank you. According to your valuable suggestion some commands and name of library functions have added in methodology section (line no: 243, 249). Also, we provided a supplementary file of R code for SHAP and Decomposition.

5) For data samples having missing values, authors should use some imputation techniques, such as SMOTE, or ADASYN.

Reply: Thank you for your comment. We also believe that exclusion is not an ideal way. We have a large data set and the variable “number of ANC visit” had 250 women’s information missing (also have missing information on other variables). That’s why we preferred to exclude these sample.

6) The authors should consider experimenting with state-of-the-art models, which may provide stronger predictive performance and deeper insights than the classical methods currently used.

Reply: We thank the reviewer for the suggestion. Accordingly, we implemented a state-of-the-art model LightGBM and CatBoost using 5-fold CV with hyperparameters. The LightGBM model achieved accuracy of 0.73, recall of 0.74, specificity of 0.72, precision of 0.83, f1-score of 0.78 and AUC of 0.79, on the other hand CatBoost achieved accuracy of 0.73, recall of 0.75, specificity of 0.69, precision of 0.82, f1-score of 0.78 and AUC of 0.79, which are comparable to the best models in our study.

Due to limitations in the library support within R 4.5.1 the additional advanced models could not be reliably implemented. We have incorporated these results of LightGBM and CatBoost into the revised manuscript and clarified this point in Methodology (line no: 285-288 & 271-273) and Result Section (line no: 565).

7) The manuscript would benefit from thorough proofreading to improve clarity, grammar, and overall readability.

Reply: Thank you so much for your comment. Considering your valuable recommendation, we revised the paper and did correction to improve readability.

Responses to Reviewer 2:

1. The Title of the Manuscript is very long. I suggest making it smaller and compact.

Reply: Thank you. As per your valuable recommendation, we proposed a new title for the manuscript, as follows (line no: 46-47):

“Determinants and disparities in skilled birth attendants during childbirth in Bangladesh: a study of machine learning and decomposition analysis”.

2. At line 199, what is MSBA, it’s not used elsewhere in paper.

Reply: Thank you. It was a mistake. Please accept our apologies. The texts are being correctly written (line no: 258 in Methodology Section).

3. The authors did not report the confusion matrix.

Reply: As per your suggestion, we have reported it on methodology part along with how they created. Also, the model metrics which were calculated from model’s confusion matrix of each model. We used test data set for predicting and creation of confusion matrix (line no: 260 & 290 in Methodology section).

4. Since the accuracy is almost nearby for all models. Whether Hyperparameter tuning, cross validation is performed or not?

Reply: Thank you for your suggestion. The authors had taken this recommendation seriously. In past default model structure was used. Now we updated the results by tuning best hyperparameter using 5-fold cross validation. And we used random search for finding best hyperparameters (line no: 258-261 in Methodology Section). The results have shown in table 4 (line no: 565 in Results Section).

5. The authors did not report about parameters choice, overfitting or underfitting of model.

Reply: Thank you. In past default model structure was used. Now we updated the results by tuning best hyperparameter using 5-fold cross validation. And we used random search for finding the best hyperparameters. We justified the performance of model based on AUC value for avoiding overfitting and underfitting of model (line no: 297-299 in Methodology Section). The models performances reported in Table 4 (line no: 565-566 in Results Section).

6. What were the hyperparameters and how were they tuned?

Reply: Thank you for your comment. In past default model structure was used. Now we updated the results by tuning best hyperparameter using 5-fold cross validation. And we used random search for hyperparameters with a tune length of 10 (line no: 258-261 in Methodology Section). The results have shown in table 4 (line no: 565 in Results Section).

7. It is unclear whether performance differences between models are statistically significant.

Reply: As per the recommendation of another respected reviewer, we performed DeLong’s test to find out performance difference in between models (line no: 298-299 in Methodology Section). The results have shown in Table 5 (line no: 595 in Results Section).

8. It’s not mentioned that after applying SMOTE how many cases are considered for further evaluation. Also, When SMOTE has applied after or before split?

Reply: Thank you for your comment. We had applied SMOTE to balance the training dataset only and it was applied after splitting the dataset (Details discussed on Methodology section; line no: 230-233).

9. A Beeswarm SHAP plot can be added

Reply: Thank you. A Beeswarm SHAP plot has added according to your suggestion. (Figure 8)

10. Its better if authors can provide a research flow and Overview of the machine learning framework applied for better understanding to readers.

Reply: Thank you. Considering your valuable suggestion, a research flow and ML framework is represented in figure 1. (Addressed at Methodology section; line no: 233- 234)

11. The conclusion is little longer. It’s better to keep strengths and limitations as separate section.

Reply: As per your valuable recommendation, the authors separated the strength and limitation. (Line no: 729-747 in Discussion Section).

12. Based on this study what are the possible recommendations.

Reply: Thank you again for your valuable suggestion. Possible recommendations are added after strength and limitation as a separate section (line no: 748-773 in Discussion Section).

13. The Abstract is too long, weak Conclusion and Introduction section lacking literature review on ML and XAI applications for SBA prediction. The methodology section did not mentioned utilization of XAI.

Reply: Considering your important recommendation, the authors tried hard to find more literatures of ML application in SBA context. But there were not so many studies found who used ML and SHAP based interpretability in this particular topic. Two published literatures found on African population. (Ref 13 & 32) The Abstract (line no: 88-104) and Conclusion (line no: 805-826 in Conclusion Section) is rewritten based on your suggestion. Also, Details about application of XAI added in the Methodology part line no: 300-309.

14. For the present study, How and why the initial independent variables are selected.

Reply: Thank you for your comment. We reviewed published literatures on Skilled Birth Attendant. Based on the review we selected the independent variables which mentioned in methodology section (line no: 217 in Methodology Section).

15. A minor English editing is required.

Reply: Thank you for your suggestion. We have read the manuscript thoroughly for searching grammatical mistakes to correct those mistakes. Also, we did try to improve the readability by rewriting some texts across the whole manuscript.

---

## [Decision Letter · Decision Letter 2]

8 Mar 2026

PONE-D-25-44386R2Determinants and disparities in skilled birth attendants during childbirth in Bangladesh: a study of machine learning and decomposition analysisPLOS One

Dear Dr. Parvin,

Thank you for submitting your manuscript to PLOS ONE. After careful consideration, we feel that it has merit but does not fully meet PLOS ONE’s publication criteria as it currently stands. Therefore, we invite you to submit a revised version of the manuscript that addresses the points raised during the review process.

We look forward to receiving your revised manuscript.

Kind regards,

Md. Obaidur Rahman, Ph.D.

Academic Editor

PLOS One

Journal Requirements:

Reviewers' comments:

Reviewer's Responses to Questions

**Comments to the Author**

1. If the authors have adequately addressed your comments raised in a previous round of review and you feel that this manuscript is now acceptable for publication, you may indicate that here to bypass the “Comments to the Author” section, enter your conflict of interest statement in the “Confidential to Editor” section, and submit your "Accept" recommendation.

Reviewer #1: All comments have been addressed

Reviewer #2: All comments have been addressed

2. Is the manuscript technically sound, and do the data support the conclusions?

Reviewer #1: Yes

Reviewer #2: Yes

3. Has the statistical analysis been performed appropriately and rigorously? 

Reviewer #1: Yes

Reviewer #2: Yes

4. Have the authors made all data underlying the findings in their manuscript fully available?

Reviewer #1: Yes

Reviewer #2: Yes

5. Is the manuscript presented in an intelligible fashion and written in standard English?

Reviewer #1: Yes

Reviewer #2: Yes

6. Review Comments to the Author

Reviewer #1: The authors have addressed the key points raised in my initial review. They strengthened the ML analysis through proper hyperparameter tuning and 5-fold cross-validation, clarified the ANN architecture, and now report performance with confidence intervals and formal model comparisons using the DeLong’s test.

They expanded the interpretability analysis with additional SHAP plots and improved transparency by adding methodological details and supplementary code. While missing data are still handled via exclusion, the rationale is now clearly stated in the manuscript.

Reviewer #2: 1. I suggest retaining abbreviations in the abstract only if they are used within the abstract itself. In the main manuscript, each abbreviation should be defined at its first occurrence, after which the abbreviated form can be used consistently.

2. It would be beneficial if the authors could provide the code and data used in the study to improve the reproducibility and transparency of the analysis.

3. I suggest moving the detailed parameter information currently presented in Table 4 to the supplementary material to improve the readability of the main text. In addition, the authors should specify the software and libraries used to implement the algorithms, either in the Methods section or in the supplementary material.

4. Kindly improve the language.

5. Improve the quality of figure 4, 6, 7 and 8. Its better to keep as subplots like A and B. Title can be reduced.

7. PLOS authors have the option to publish the peer review history of their article (what does this mean?). If published, this will include your full peer review and any attached files.

Reviewer #1: **Yes:** Nafiz Imtiaz Khan

Reviewer #2: No

---

## [Author Response · Author response to Decision Letter 3]

18 Mar 2026

Manuscript Number: PONE-D-25-44386R2

EMID:77c84a23547f9504

Replies to the Reviewers’ Comments

The authors would like to thank the respected editorial manager and reviewers for providing valuable comments, constructive criticisms and useful suggestions that have greatly improved the quality, presentation and style of the manuscript. The authors have done their level best to address all the comments raised by the respected reviewers in the revised manuscript. The text edited in the revised manuscript is typed in red colour and line number are mentioned based on “Revised manuscript with track change” file.

The point-by-point replies to the respected editorial manager and reviewers’ comments are as follows:

Responses on journal requirements:

Reply: Thank you very much for this important reminder. We have carefully reviewed the reference list and corrected references 3, 9, 16, 17, 61, and 69 to ensure they comply with the required journal style. No references were added or removed during this revision.

Responses to Reviewer 1:

Reviewer #1: The authors have addressed the key points raised in my initial review. They strengthened the ML analysis through proper hyperparameter tuning and 5-fold cross-validation, clarified the ANN architecture, and now report performance with confidence intervals and formal model comparisons using the DeLong’s test.

They expanded the interpretability analysis with additional SHAP plots and improved transparency by adding methodological details and supplementary code. While missing data are still handled via exclusion, the rationale is now clearly stated in the manuscript.

Reply: Thank you for your positive and encouraging feedback. We appreciate your valuable comments and support, which helped improve the quality of our manuscript.

Responses to Reviewer 2:

Recommendations:

1. I suggest retaining abbreviations in the abstract only if they are used within the abstract itself. In the main manuscript, each abbreviation should be defined at its first occurrence, after which the abbreviated form can be used consistently.

Reply: Thank you for your valuable suggestion. We have corrected the use of abbreviation as you recommended (line no 28 to 32 in Abstract) (line no 126, 128, 189, 194, 206, 226, 229, 357, 462 in main texts).

2. It would be beneficial if the authors could provide the code and data used in the study to improve the reproducibility and transparency of the analysis.

Reply: Thank you for your valuable suggestion. To enhance the reproducibility and transparency of our analysis, we have now provided the data, STATA and R code used in our paper on google drive. The data and additional data file for map creation is also available on the following google drive link:

https://drive.google.com/drive/folders/1eEoIIB3x62POgJvWKMgL8z3pXa2HqgOB?usp=sharing

3. I suggest moving the detailed parameter information currently presented in Table 4 to the supplementary material to improve the readability of the main text. In addition, the authors should specify the software and libraries used to implement the algorithms, either in the Methods section or in the supplementary material.

Reply: Thank you. Considering your valuable suggestion, we moved the parameter information to supplementary file along with specifying the software (R version 4.5.1) and library names in a separate column (Supplementary Table S2).

4. Kindly improve the language.

Reply: Thank you for your valuable suggestion. We have carefully reviewed the manuscript and made the necessary revisions accordingly to improve the overall language and clarity.

5. Improve the quality of figure 4, 6, 7 and 8. Its better to keep as subplots like A and B. Title can be reduced.

Reply: Thank you for your valuable recommendation. As per your recommendation we have improved the quality of figure 4 and 6 and we made figure 7 and 8 as subplots (a) and (b). Also, title of the figures 4, 7 and 8 have been reduced (line no 548, 568, 588-589, 603-607 in Results section).

---

## [Editor Report · Decision Letter 3]

23 Mar 2026

Determinants and disparities in skilled birth attendants during childbirth in Bangladesh: a study of machine learning and decomposition analysis

PONE-D-25-44386R3

Dear Dr. Parvin,

We’re pleased to inform you that your manuscript has been judged scientifically suitable for publication and will be formally accepted for publication once it meets all outstanding technical requirements.

Kind regards,

Md. Obaidur Rahman, Ph.D.

Academic Editor

PLOS One
---

## [Editor Report · Acceptance letter]

PONE-D-25-44386R3

PLOS One

Dear Dr. Parvin,

I'm pleased to inform you that your manuscript has been deemed suitable for publication in PLOS One. Congratulations! Your manuscript is now being handed over to our production team.

Kind regards,

on behalf of

Dr. Md. Obaidur Rahman

Academic Editor

PLOS One